# LONG CHAIN-OF-THOUGHT REASONING ACROSS LANGUAGES

**Josh Barua    Seun Eisape    Kayo Yin    Alane Suhr**
University of California, Berkeley
{joshbarua,suhr}@berkeley.edu

## ABSTRACT

While large reasoning models have shown remarkable ability to generate long chains-of-thought (CoTs) in English, we still lack understanding of how these long-form reasoning abilities transfer to the vast majority of the world's languages. In this work, we systematically investigate four key stages of model development–scaling, pretraining, post-training, and inference–to understand how long CoT capabilities extend beyond English. We compare two reasoning settings across nine non-English target languages: En-CoT, where models process target-language inputs, but reason in English; and Target-CoT, where models both process inputs and generate long CoTs in the target language. We find that scaling reasoning model size improves multilingual task performance in En-CoT, but Target-CoT performance lags behind. This gap widens for tasks requiring long, multi-step CoTs such as mathematical reasoning. Shifting to pretraining, we find that adding a specialized reasoning stage enhances En-CoT performance but degrades Target-CoT, whereas broad multilingual pretraining improves both modes simultaneously. Given the scarcity of high-quality reasoning traces in languages other than English, we explore synthetic data curation approaches for post-training. We demonstrate that fine-tuning on reasoning traces automatically translated from gold English traces outperforms fine-tuning on target-language traces distilled from large reasoning models. Finally, we report disparities in inference efficiency between languages and uncover language-specific failure modes in CoTs. We release models, datasets, and code to foster further research. [1]

## 1    INTRODUCTION

Reasoning has emerged as a central focus of large language model (LLM) research in recent years. By scaling inference-time compute to generate chains-of-thought before answering queries, LLMs have achieved expert-level performance in coding, mathematics, and scientific domains (OpenAI, 2024; DeepSeek-AI et al., 2025). These long chains-of-thought demonstrate sophisticated reasoning processes—exploring different solution pathways, identifying and correcting mistakes, and setting intermediate subgoals—which result in substantially longer reasoning traces. They not only improve accuracy but also expose intermediate steps, offering transparency, error-recovery, and interpretability (Korbak et al., 2025).

Despite this progress, reasoning research has focused almost exclusively on English. Even when models are evaluated on multilingual tasks, they often generate the intermediate reasoning in English rather than matching the input language (Qi et al., 2025). This creates both a scientific and practical blind spot. From a scientific perspective, it is unknown whether the mechanisms of long-form reasoning generalize and transfer to other languages, or whether they rely on English-centric pretraining signals. From a practical perspective, producing reasoning steps in English reduces usability for billions of non-English speakers: it becomes harder to audit steps, reproduce a solution, or diagnose errors when the explanation is not in the user's language. It also potentially limits reasoning quality in tasks that are culturally or linguistically grounded.

Therefore, evaluating reasoning models requires assessing two distinct capabilities: their ability to understand inputs in languages other than English, and their capacity to accurately reason in a user's native language. In this work, we present a systematic study of long chain-of-thought reasoning across nine languages beyond English: three high, three mid and three low-resource. We systematically compare

---

[1]https://github.com/Berkeley-NLP/Multilingual-Long-CoT

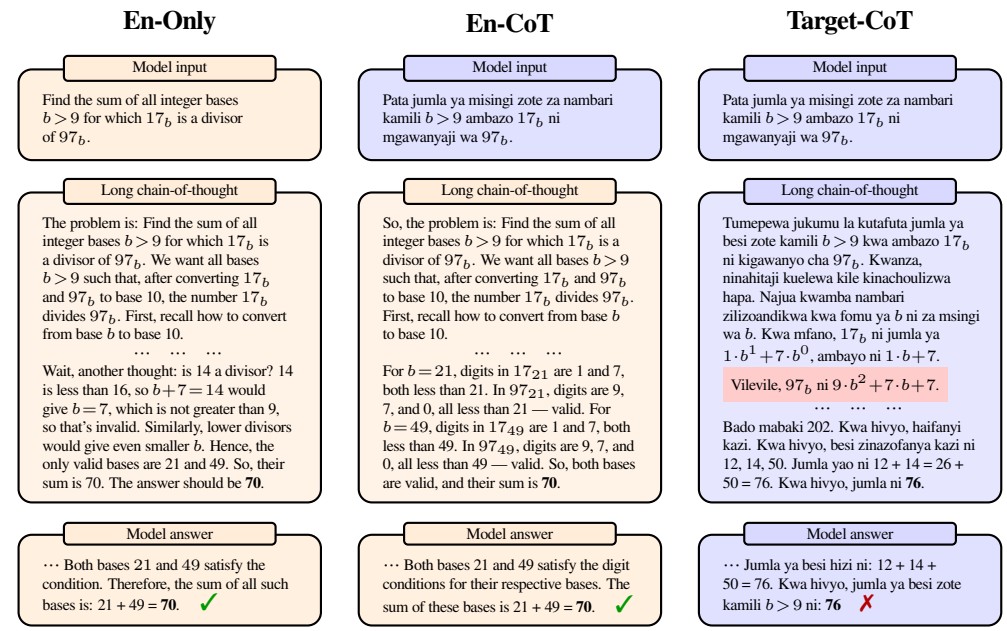

Figure 1: **Example inputs, long chains-of-thought, and answers for *En-Only*, *En-CoT*, and *Target-CoT* settings,** drawn from DeepSeek-R1-Distill-Llama-70B on AIME 2025. Orange boxes denote English text and blue boxes denote Swahili text. En-Only (input & reasoning in English) and En-CoT (input in Swahili, reasoning in English) lead to correct answers, while Target-CoT (input & reasoning in Swahili) contains a reasoning error (highlighted in red) and leads to an incorrect answer.

three reasoning setups (illustrated in Figure 1): En-Only, where models process input and reason in English; En-CoT, where models process target-language inputs but reason in English; and Target-CoT, where models perform both processing and reasoning in the target language. We identify weaknesses and opportunities for improving multilingual LLMs across four key dimensions of model development: scaling, pretraining, post-training, and inference. We provide empirical findings on mathematical and general-domain reasoning, highlighting where advances in English reasoning fail to generalize, and where targeted multilingual interventions close the gap.

In particular, our study reveals that while model scaling and multilingual pretraining substantially improve cross-lingual comprehension, performance in target-language chain-of-thought remains far behind English even at large parameter scales. Moreover, lightweight post-training using translated reasoning traces lifts multilingual performance far more effectively than large-scale English post-training, enabling competitive gains for mid- and low-resource languages with modest data. Finally, inference-time analysis highlights distinct failure modes when reasoning in non-English languages, underscoring the challenges unique to multilingual reasoning and paving the way for further improvements.

## 2 RELATED WORK

**Long CoT Reasoning.** Early studies demonstrated that inference-time techniques enhance LLM reasoning capabilities by enabling models to perform intermediate reasoning steps before generating final answers. These techniques include chain-of-thought prompting (Wei et al., 2022), which guides models to explicitly show their reasoning process, and scratchpads (Nye et al., 2021), which provide dedicated spaces for intermediate computations. By scaling inference-time compute through longer CoTs, frontier LLMs have achieved expert-level performance on challenging benchmarks across programming and STEM disciplines (OpenAI, 2024; DeepSeek-AI et al., 2025). Recent work has sought to understand inference-time scaling (Wu et al., 2024; Snell et al., 2025), with special attention given to long CoT reasoning as a scaling approach (Yeo et al., 2025; Muennighoff et al., 2025).

**Multilingual Reasoning.** Early research on multilingual reasoning in LLMs found that English CoT outperforms target language reasoning on grade-school math problems (Cobbe et al., 2021; Shi et al., 2023).

This finding motivated a series of works that sought to bridge the multilingual gap by using English as a pivot language when reasoning. Researchers have explored this approach both externally, by translating non-English queries into English before processing (Huang et al., 2023; Qin et al., 2023; Ko et al., 2025), and internally, by aligning the model's latent representations across languages to an English-centric space (She et al., 2024; Huang et al., 2024; Yoon et al., 2024; Ruan et al., 2025).

These works were conducted in the short CoT setting, where reasoning chains are typically a few steps. Recent studies have explored whether more advanced inference-time scaling techniques, including long CoT, transfer across languages. Yong et al. (2025) demonstrate that scaling inference compute for models fine-tuned on English reasoning traces improves mathematical reasoning in high-resource languages but not low-resource languages. Son et al. (2025) find that inference-time scaling methods such as budget forcing are comparable to traditional scaling methods like best-of-N once constrained to similar levels of inference FLOPs. These studies, however, share a key limitation: they focus primarily on models that perform reasoning in English. Only recently has research begun examining scenarios where models reason directly in non-English languages. Qi et al. (2025) identify a trade-off between language compliance and accuracy, while Wang et al. (2025) explore the phenomenon of language mixing during reasoning. Our work adopts a complementary perspective by exploring the cross-lingual transfer of long CoT reasoning through all stages of the model development pipeline.

## 3 EXPERIMENTAL FRAMEWORK

**Preliminaries on Long CoT reasoning.** We use the term *long chain-of-thought (long CoT)* to refer to sequences of thinking tokens that extend far beyond typical model responses–often tens of thousands of tokens–and demonstrate complex behaviors such as branching, backtracking, and self-verification (Gandhi et al., 2025). Training models for long CoT typically involves two stages: supervised fine-tuning (SFT) on datasets of questions paired with reasoning traces followed by reinforcement learning (RL) that rewards correct final answers using techniques like PPO (Schulman et al., 2017). However, developing long CoT capabilities in non-English languages presents significant challenges. High-quality reasoning traces for SFT are scarce outside of English, and RL-based reward shaping is highly sensitive in non-English contexts, leading to poor language compliance (Park et al., 2025).

**Reasoning setup.** Following Ko et al. (2025), we decompose a model's performance on multilingual reasoning tasks into two distinct factors: understanding the input language ($L_{\text{input}}$) and reasoning in a particular language ($L_{\text{reason}}$). Throughout this paper, we refer to the language of the input – and by extension, the user's desired language for the interaction – as the *target language*. To isolate the influence of target-language comprehension from reasoning capabilities in task success, we consider three experimental settings (see Figure 1). **En-Only** refers to having English as both input and reasoning languages, and showcases the English baseline performance. In **En-CoT**, the model receives inputs in the target language but reasons in English. In **Target-CoT**, both input and reasoning are in the target language. Although target-language solutions would be ideal, we instruct models in the En-CoT setting to output solutions in English due to validation challenges—short non-English texts with mathematical notation often cause inconsistent language detection. The En-CoT setting evaluates **comprehension**: weak performance compared to En-Only indicates difficulty understanding target-language inputs. The Target-CoT setting evaluates **reasoning**: comparing its accuracy to En-CoT reveals the model's ability to reason in the target language.

**Languages.** On top of English [EN], we study three high-resource languages (Chinese [ZH], French [FR], Japanese [JA]), three mid-resource languages (Afrikaans [AF], Thai [TH], Latvian [LV]), and three low-resource languages (Marathi [MR], Telugu [TE], Swahili [SW]). We assign resource levels based on each language's representation in the mC4 pretraining dataset (Xue et al., 2021). These languages cover a broad mix of scripts, geographic regions, and language families, allowing us to study cross-lingual CoT reasoning at scale (see Appendix A.1 for more details on each language).

**Multilingual benchmarks.** We perform evaluation on two types of tasks: mathematical problem solving and multi-task language understanding. For math, we use MATH-500 (Hendrycks et al., 2021) and AIME-Combined (concatenating AIME 2024 and 2025 (Art of Problem Solving, 2025)). We automatically translate MATH-500 and AIME 2025 problems using Gemini-2.0-Flash and use the MCLM benchmark (Son et al., 2025) to source translated versions of AIME 2024 problems. For general reasoning, we use MMLU-ProX (Xuan et al., 2025), a multilingual, multi-disciplinary benchmark that tests knowledge-grounded reasoning beyond math. It covers all our languages of study except Latvian. Additional details on inference and evaluation can be found in Appendix A.5.

**Experiments.** In Section 4, we report the effects of scaling model size, separating gains in target language comprehension from target language reasoning. In Section 5, holding scale and post-training fixed, we compare pretrained backbones to disentangle the effects of a reasoning-specialized stage versus broad multilingual coverage. In Section 6, we examine post-training, fine-tuning with small language-specific datasets created either by translating English traces or by distilling target language traces. Finally, we analyze inference behavior in Section 7, quantifying efficiency–accuracy trade-offs and cataloging language-specific failure modes.

## 4    SCALING MODEL PARAMETERS

Prior work has consistently shown that scaling model parameters improves multilingual performance across various tasks (Conneau et al., 2020; Li et al., 2021; Chowdhery et al., 2023; He et al., 2025). We revisit this question in the context of long CoT reasoning. An ideal scaling experiment should (i) keep the post-training recipe and dataset fixed, (ii) use a common pretrained backbone to reduce corpus/tokenizer differences, and (iii) span multiple sizes to reveal scaling trends. We meet these criteria with the DeepSeek-R1-Distill series: the 1.5B–32B variants share the same post-training procedure on the same 800k reasoning traces and derive from the same base family (Qwen 2.5), allowing us to attribute changes in performance primarily to parameter count. Increasing model capacity through scaling may not uniformly improve input comprehension and reasoning in the target language. Thus, we report performance in both En-CoT (comprehension) and Target-CoT (comprehension + reasoning) to pinpoint where scaling provides the most benefit.

**En-CoT performance.** We first examine how increasing model size affects input comprehension in the target language. Figure 2 reports En-CoT results, where models receive inputs in the target language but perform reasoning in English. The scaling trends are language-dependent. For high-resource languages such as Chinese and French, all model sizes (1.5B, 7B, 14B, and 32B) perform on par with the En-Only baselines (dashed horizontal lines), suggesting that target-language comprehension is already strong even in smaller models. For Japanese, Afrikaans, Thai, and Latvian, the 7B models are closest to the En-Only baselines, and larger models remain within roughly 10% of En-Only performance. This relatively small gap indicates that, once pretraining exposure is sufficient, scaling model capacity largely removes comprehension as a barrier to cross-lingual reasoning. In contrast, lower-resource languages (Marathi, Telugu, Swahili) remain substantially below the En-Only baselines at every scale, including 32B. This persistent deficit highlights that English reasoning does not universally solve cross-lingual long CoT, because comprehension bottlenecks can still dominate for languages with limited pretraining coverage.

**Target-CoT performance.** As shown in Figure 2, reasoning in the target language (Target-CoT) never approaches English reasoning capabilities, even for high-resource languages like Chinese and French. Remarkably, at 32B parameters, all tested languages perform at or below the 7B En-Only baseline—a model with 4× fewer parameters. Lower-resource languages (Marathi, Telugu, Swahili) show virtually no sensitivity to scale, with performance remaining near zero. Overall, switching from English to target language reasoning at 32B reduces accuracy by 28.8% on average. These results reveal that while scaling effectively addresses input comprehension challenges, it fails to enable sustained, structured reasoning in non-English languages. This contrasts with findings on short CoT, where performance gaps stem primarily from input comprehension rather than the language used for reasoning (Ko et al., 2025). In Appendix B.2, we provide evidence that the length of the reasoning chain explains this discrepancy and observe cases where Target-CoT *outperforms* En-CoT.

## 5    MULTILINGUAL PRETRAINING

Recent technical reports indicate that companies are revisiting pretraining for reasoning, adding dedicated "reasoning stages" (Yang et al., 2025a; GLM 4.5 Team et al., 2025) to improve foundational capabilities. Yet the relative contributions of broad multilingual coverage versus domain-specialized reasoning data to downstream multilingual reasoning remain underexplored. We take a first step toward disentangling these factors through a controlled study that fixes model scale and post-training while varying only the pretrained backbone.

**Models.** Qwen 2.5-7B provides a strong multilingual base trained on 18T tokens across 29 languages, including coverage of Chinese, French, Japanese, and Thai (Yang et al., 2025b). Building on this base, Qwen2.5-Math-7B (Yang et al., 2024) continues pretraining with over 1T tokens of math-focused data in English and Chinese, enabling a controlled test of domain-specialized pretraining when multilingual

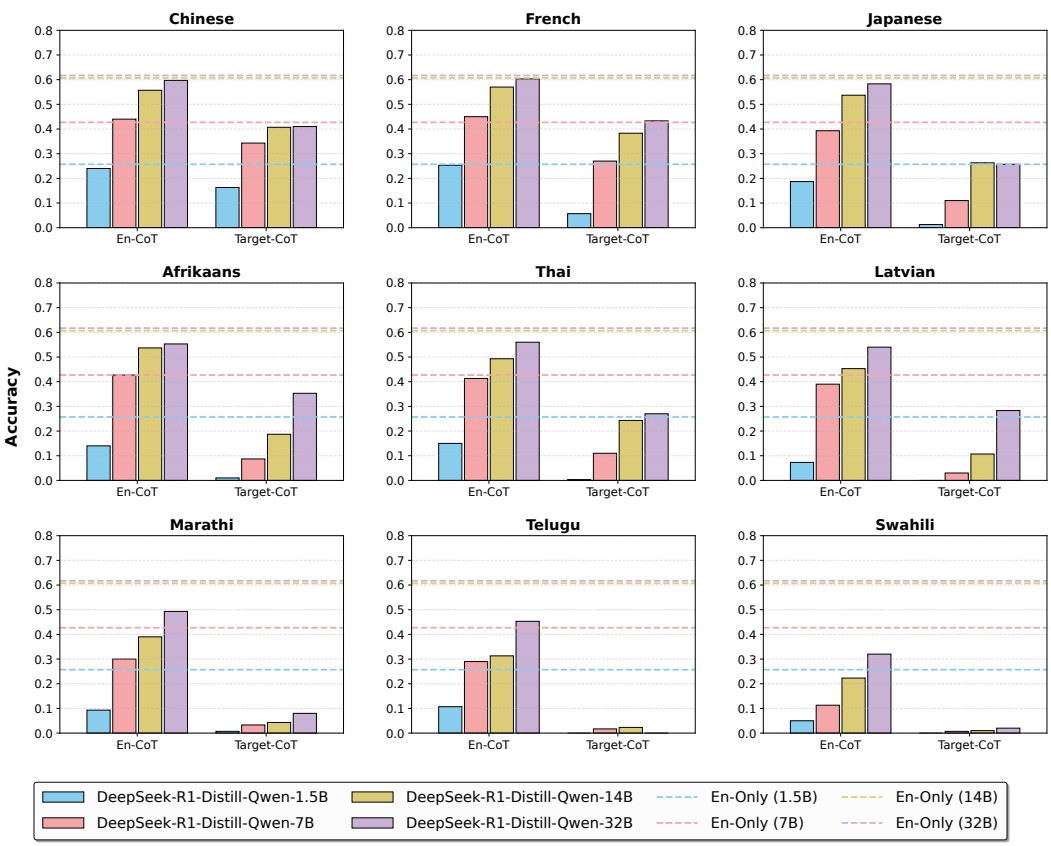

Figure 2: **Evaluation of scaling trends in DeepSeek-R1-Distill models on AIME-Combined.** For high- and mid-resource languages, En-CoT performance increases and approaches En-Only performance with scale, while Target-CoT performance is consistently lower than En-CoT, highlighting target-language reasoning as the bottleneck.

coverage is held constant. To examine how broad multilingual coverage influences multilingual reasoning, we select two additional models. Qwen3-8B-Base[2] is pretrained on 36T tokens across 119 languages and dialects, with coverage for all tested languages (Yang et al., 2025a). Gemma3-12B-PT (Gemma Team et al., 2025) is pretrained on 12T tokens covering over 140 languages.

**Experimental setup.** We perform supervised fine-tuning of all models on 20k English reasoning traces randomly sampled from OpenThoughts3-1.2M, which encompasses math, code, and science questions (Guha et al., 2025) (see Appendix A.3 for complete details). We only evaluate on mathematical reasoning (MATH-500, AIME-Combined) tasks to align with the mathematical pretraining of Qwen2.5-Math-7B.

**Impact of specialized reasoning pretraining.** We compare Qwen2.5-7B with Qwen2.5-Math-7B (Table 1) to measure the impact of specialized reasoning pretraining. In En-CoT on MATH-500, most languages see small lifts, though Swahili declines by 5.4%. By contrast, Target-CoT drops sharply for many languages (French -46.0%, Afrikaans -39.4%), suggesting weakened target language reasoning despite in-domain content. Surprisingly, this degradation occurs even in Chinese (-36.4%), despite Chinese data being included in the reasoning pretraining stage. AIME-Combined shows the same asymmetry: En-CoT improves by 5.6% on average while Target-CoT falls by 4.3%. Hence, specialized pretraining can improve English reasoning within its domain but undermines target language performance, even for languages explicitly included in the specialized training stage.

**Impact of multilingual pretraining.** Comparing scores between the Qwen 2.5 models and Qwen 3 and Gemma 3 is challenging due to differences in pretraining, architecture, and scale. We therefore

---

[2]Qwen uses a new naming convention for base models in the Qwen 3 series, hence the discrepancy.

Table 1: **Evaluation of each base model after identical English post-training.** AVG excludes English, and EPR (English Performance Recovered) is computed as AVG / EN (%), measuring cross-lingual transfer efficiency. Pretraining on math data improves En-CoT performance but degrades Target-CoT. Broad multilingual pretraining in Qwen3-8B-Base and Gemma3-12B-PT recover 85-97% of English performance for En-CoT while also yielding substantial EPR gains for Target-CoT.

| | Base Model | EN | ZH | FR | JA | AF | TH | LV | MR | TE | SW | AVG | EPR |
|---|---|---|---|---|---|---|---|---|---|---|---|---|---|
| | **MATH-500** | | | | | | | | | | | | |
| En-CoT | Qwen2.5-7B | 90.2 | 89.6 | 89.6 | 87.0 | 88.6 | 87.0 | 82.2 | 74.8 | 74.2 | 50.4 | 80.4 | 89.1 |
| | Qwen2.5-Math-7B | 92.2 | 91.4 | 90.6 | 89.2 | 89.6 | 88.6 | 81.8 | 76.0 | 80.8 | 45.0 | 81.4 | 88.3 |
| | Qwen3-8B-Base | 94.6 | 96.0 | 94.4 | 93.2 | 93.4 | 91.6 | 93.6 | 89.8 | 90.2 | 71.4 | 90.4 | 95.6 |
| | Gemma3-12B-PT | 76.6 | 75.8 | 77.8 | 76.4 | 75.2 | 73.8 | 74.6 | 75.6 | 72.8 | 67.0 | 74.4 | **97.1** |
| Target-CoT | Qwen2.5-7B | 90.2 | 76.2 | 67.2 | 33.4 | 45.0 | 28.6 | 19.4 | 11.2 | 9.4 | 0.2 | 32.3 | 35.8 |
| | Qwen2.5-Math-7B | 92.2 | 39.8 | 21.2 | 21.6 | 5.6 | 11.6 | 6.4 | 3.2 | 3.6 | 0.2 | 12.6 | 13.7 |
| | Qwen3-8B-Base | 94.6 | 70.6 | 89.2 | 77.6 | 76.8 | 77.4 | 70.0 | 45.6 | 30.0 | 21.2 | 62.0 | **65.6** |
| | Gemma3-12B-PT | 76.6 | 41.6 | 55.4 | 32.8 | 60.4 | 34.4 | 52.8 | 23.0 | 16.2 | 40.4 | 39.7 | 51.8 |
| | **AIME-Combined** | | | | | | | | | | | | |
| En-CoT | Qwen2.5-7B | 30.0 | 29.7 | 28.3 | 26.0 | 29.0 | 27.3 | 25.7 | 21.0 | 16.7 | 10.0 | 23.7 | 79.0 |
| | Qwen2.5-Math-7B | 36.7 | 31.3 | 38.0 | 33.7 | 36.3 | 30.7 | 33.3 | 25.3 | 27.0 | 8.3 | 29.3 | 79.8 |
| | Qwen3-8B-Base | 42.3 | 43.0 | 45.7 | 40.0 | 46.0 | 44.0 | 41.7 | 42.0 | 34.0 | 24.7 | 40.1 | **94.8** |
| | Gemma3-12B-PT | 14.3 | 12.0 | 11.7 | 10.3 | 13.7 | 11.0 | 12.3 | 13.3 | 14.0 | 11.3 | 12.2 | 85.2 |
| Target-CoT | Qwen2.5-7B | 30.0 | 21.3 | 15.7 | 8.0 | 6.0 | 2.3 | 0.7 | 0.0 | 0.0 | 0.0 | 6.0 | 20.0 |
| | Qwen2.5-Math-7B | 36.7 | 9.3 | 3.0 | 1.0 | 1.3 | 0.3 | 0.7 | 0.0 | 0.0 | 0.0 | 1.7 | 4.6 |
| | Qwen3-8B-Base | 42.3 | 24.3 | 31.0 | 21.3 | 23.3 | 22.3 | 12.7 | 5.0 | 3.0 | 0.3 | 15.9 | **37.6** |
| | Gemma3-12B-PT | 14.3 | 4.0 | 9.0 | 3.0 | 7.7 | 4.7 | 6.3 | 3.3 | 2.0 | 5.7 | 5.1 | 35.5 |

focus on the percentage of English performance recovered (EPR) for each model. Broad multilingual pretraining in Qwen3-8B-Base and Gemma3-12B-PT nearly closes the target language comprehension gap. In En-CoT, both models recover at least 95% of En-Only performance on MATH-500 and 85-95% on AIME-Combined (Table 1), with shortfalls concentrated in the lowest-resource cases. However, when reasoning in the target language, large gaps remain with EPR ranging from 35% to 65%. This pattern indicates that multilingual pretraining establishes the foundation for understanding non-English inputs, but further work is needed to teach models to produce long, structured CoTs in those languages. Importantly, the EPR of Qwen 3 and Gemma 3 are substantially higher than the Qwen 2.5 models, demonstrating the advantage of broad language coverage in pretraining for more balanced reasoning.

# 6 SYNTHETIC DATA FOR POST-TRAINING

In Section 5, we found that multilingual pretraining helps bridge comprehension gaps between languages, yet the ability to generate long reasoning chains beyond English remains brittle. This section explores whether post-training can meaningfully improve target-language reasoning. A fundamental challenge immediately emerges: high-quality reasoning traces are rarely documented in languages beyond English and Chinese. This scarcity raises a critical question—how should we optimally curate synthetic data for multilingual reasoning? We identify two primary approaches. **Translation-based approach:** translate existing English reasoning traces into target languages, leveraging the extensive availability of English data. **Direct distillation approach:** employ language forcing to directly distill target-language traces from advanced reasoning models like DeepSeek-R1, potentially preserving language-specific reasoning patterns. In this section, we investigate both the effectiveness of post-training for target-language reasoning and compare these two synthetic data curation strategies.

## 6.1 EXPERIMENTAL SETUP

We curate two synthetic datasets from s1k (Muennighoff et al., 2025), a dataset of 1,000 high-quality English reasoning traces distilled from DeepSeek-R1. **Translated-s1k**: a translated version using Gemini-2.0-Flash. We selected Gemini-2.0-Flash after verifying that it outperforms strong machine translation models on the FLORES-200 benchmark and achieves higher quality estimation scores when translating reasoning traces (see Appendix B.6 for details). **Distilled-s1k**: directly distilled traces from DeepSeek-R1, using language forcing to elicit reasoning in target languages. We continue re-sampling until all 1,000 distilled traces for each language pass our language compliance check (Appendix A.7). For each dataset and language combination, we fine-tune separate Qwen3-8B-Base models, resulting in 18 models trained exclusively on target-language supervision. Appendix A.3 provides detailed hyperparameter configurations for training. We evaluate on both mathematical (MATH-500, AIME-Combined) and general (MMLU-ProX) reasoning tasks. As a baseline, we include Qwen3-8B-Base fine-tuned on 20k English traces from OpenThoughts3 (best-performing model in Section 5). Although this introduces a difference in compute, this setup represents a realistic tradeoff practitioners face: choosing between abundant English data sources and smaller target-language datasets. In Appendix B.4, we present results under compute-equivalent conditions. For reference, we also include results from Qwen3-8B, a state-of-the-art reasoning model at this scale.

## 6.2 RESULTS

**Translation vs Distillation.** In Table 2, translation emerges as the generally stronger and more stable approach over distillation, achieving particularly substantial gains in Chinese (+24.2%) and French (+9.2%) when averaged across the three benchmarks. Most other languages show more modest improvements of 1-4% on average, with Marathi being the sole exception where Distilled-s1k outperforms its counterpart. For mid- and low-resource languages, the performance gap between the two approaches remains relatively narrow, suggesting that practitioners could reasonably choose either method based on available resources. However, several open questions remain. First, how do these approaches scale with increased data? Additionally, as next-generation reasoning models develop stronger multilingual capabilities, the relative effectiveness of distillation methods may evolve, warranting continued evaluation of both strategies.

**When to Use English vs Target-language Data.** Given the scarcity of target-language reasoning data, practitioners must choose between leveraging abundant English resources or investing in smaller target-language datasets. Our experiments reveal that this choice depends on the target language's resource level. For high-resource languages like French and Japanese, fine-tuning on OpenThoughts3-20k (English-only) achieves modest improvements over both Translated-s1k and Distilled-s1k variants, suggesting these languages can leverage cross-lingual transfer from English reasoning data. In contrast, mid- and low-resource languages show substantial gains from target-language fine-tuning, achieving competitive performance with just 1,000 traces—20× less data than the English baseline. Remarkably, this minimal fine-tuning even enables our models to surpass Qwen3-8B across Marathi, Telugu, and Swahili. These findings offer clear practical guidance: while target-language synthetic data benefits all languages, it becomes essential for mid- and low-resource languages.

We provide additional validation in Appendix B.4 under compute-equivalent conditions and in Appendix B.5 using Olmo3-7B as the base model.

## 7 ANALYSIS OF LONG CoT INFERENCE

In this section, we examine the chains-of-thought themselves to identify weaknesses that benchmark scores alone cannot reveal. Our analysis focuses on two key areas: inference efficiency trends and error patterns across languages.

**Inference Efficiency.** Section 6 showed that fine-tuning on target-language data improves Target-CoT performance over English data, but the inference efficiency of these two approaches remains unclear. When generating long CoTs spanning tens of thousands of tokens, cost can be as important as accuracy in deployment. Drawing inspiration from a similar analysis done by Yong et al. (2025), we plot accuracy against average response tokens for models fine-tuned on OpenThoughts3-20k and Translated-s1k (Figure 3). We observe strong negative correlations between cost and performance in both settings ($r = -0.824$ and $r = -0.915$). We also find interesting trends within each setting. The model trained on English-only data is significantly more efficient in English, yet these gains do not transfer to other languages. In the left plot, English lies on the efficiency frontier and is far from the best-fit line that characterizes the other

Table 2: **Evaluation after fine-tuning Qwen3-8B-Base on different synthetic datasets.** All evaluations are done in the Target-CoT setting. Subscripts are calculated using OpenThoughts3-20k as the baseline for each benchmark. Qwen3-8B is a state-of-the-art reasoning model, evaluated with thinking mode enabled for long CoT. Gray indicates the absolute difference is less than or equal to 2 points. The AVG excludes English and ignores missing entries ('–'). Fine-tuning with translated data leads to higher performance than distilled data, and target-language data outperforms the English baseline, despite using 20× less data.

| SFT Dataset | EN | ZH | FR | JA | AF | TH | LV | MR | TE | SW | AVG |
|---|---|---|---|---|---|---|---|---|---|---|---|
| **MATH-500** | | | | | | | | | | | |
| OpenThoughts3-20k | 94.6 | 75.8 | 89.2 | 77.6 | 76.8 | 77.4 | 70.0 | 45.6 | 30.0 | 21.2 | 62.6 |
| Translated-s1k | $92.6_{-2.0}$ | $87.2_{+11.4}$ | $87.0_{-2.2}$ | $70.0_{-7.6}$ | $87.8_{+11.0}$ | $81.6_{+4.2}$ | $83.2_{+13.2}$ | $70.8_{+25.2}$ | $72.4_{+42.4}$ | $64.4_{+43.2}$ | 78.3 |
| Distilled-s1k | $92.6_{-2.0}$ | $60.0_{-15.8}$ | $78.8_{-10.4}$ | $64.4_{-13.2}$ | $85.4_{+8.6}$ | $83.8_{+6.4}$ | $82.0_{+12.0}$ | $76.2_{+30.6}$ | $67.2_{+37.2}$ | $57.8_{+36.6}$ | 72.8 |
| Qwen3-8B (Thinking) | 97.8 | 94.0 | 92.0 | 89.4 | 91.4 | 90.2 | 85.2 | 49.0 | 47.4 | 8.4 | 71.9 |
| **AIME-Combined** | | | | | | | | | | | |
| OpenThoughts3-20k | 42.3 | 27.0 | 31.0 | 21.3 | 23.3 | 22.3 | 12.7 | 5.0 | 3.3 | 0.3 | 16.2 |
| Translated-s1k | $33.0_{-9.3}$ | $29.3_{+2.3}$ | $29.0_{-2.0}$ | $16.0_{-5.3}$ | $27.7_{+4.4}$ | $19.3_{-3.0}$ | $24.7_{+12.0}$ | $14.7_{+9.7}$ | $15.3_{+12.0}$ | $11.3_{+11.0}$ | 20.8 |
| Distilled-s1k | $33.0_{-9.3}$ | $10.7_{-16.3}$ | $21.0_{-10.0}$ | $13.0_{-8.3}$ | $28.3_{+5.0}$ | $19.7_{-2.6}$ | $22.0_{+9.3}$ | $16.3_{+11.3}$ | $13.7_{+10.4}$ | $12.7_{+12.4}$ | 17.5 |
| Qwen3-8B (Thinking) | 72.3 | 54.3 | 49.3 | 32.7 | 38.0 | 37.3 | 26.0 | 4.3 | 4.3 | 1.0 | 27.5 |
| **MMLU-ProX** | | | | | | | | | | | |
| OpenThoughts3-20k | 71.1 | 59.4 | 66.2 | 53.9 | 57.1 | 52.9 | – | 31.5 | 15.5 | 13.9 | 43.8 |
| Translated-s1k | $69.0_{-2.1}$ | $58.7_{-0.7}$ | $59.5_{-6.7}$ | $41.2_{-12.7}$ | $55.1_{-2.0}$ | $49.1_{-3.8}$ | – | $40.0_{+8.5}$ | $34.0_{+18.5}$ | $26.5_{+12.6}$ | 45.5 |
| Distilled-s1k | $69.0_{-2.1}$ | $32.0_{-27.4}$ | $48.0_{-18.2}$ | $38.8_{-15.1}$ | $53.2_{-3.9}$ | $48.5_{-4.4}$ | – | $41.7_{+10.2}$ | $35.9_{+20.4}$ | $20.4_{+6.5}$ | 39.8 |
| Qwen3-8B (Thinking) | 78.1 | 64.8 | 67.5 | 55.4 | 59.0 | 58.2 | – | 32.0 | 30.8 | 1.5 | 46.2 |

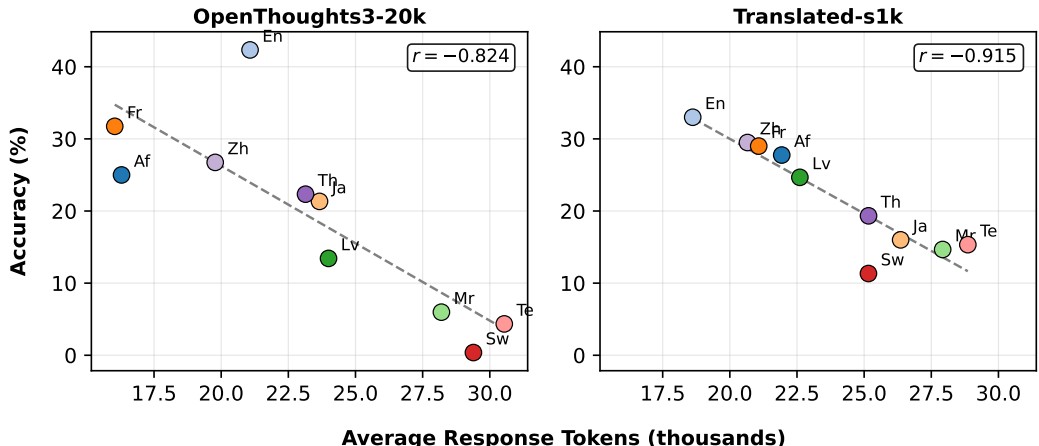

Figure 3: **Inference efficiency of fine-tuned models on AIME-Combined in terms of tokens.** Accuracy is negatively correlated with cost. Fine-tuning on translated data (right) narrows the efficiency gap across languages.

languages. In contrast, fine-tuning models on target-language data narrows the efficiency gap across languages (right plot). This approach benefits low-resource languages, though it may reduce efficiency in high-resource languages that could otherwise leverage cross-lingual transfer from English. While token counts directly relate to computational costs, they may not fully capture information content due to tokenization disparities across languages (Petrov et al., 2023). In Appendix B.7, we use bytes instead of tokens and observe weaker negative correlations ($r = -0.155$ and $r = -0.355$), suggesting that raw information content and performance are less related.

**Error Analysis.** Complex problems requiring long CoTs manifest distinct error patterns across languages. Using our taxonomy of five error categories (described in Appendix A.8), we analyzed incorrect responses

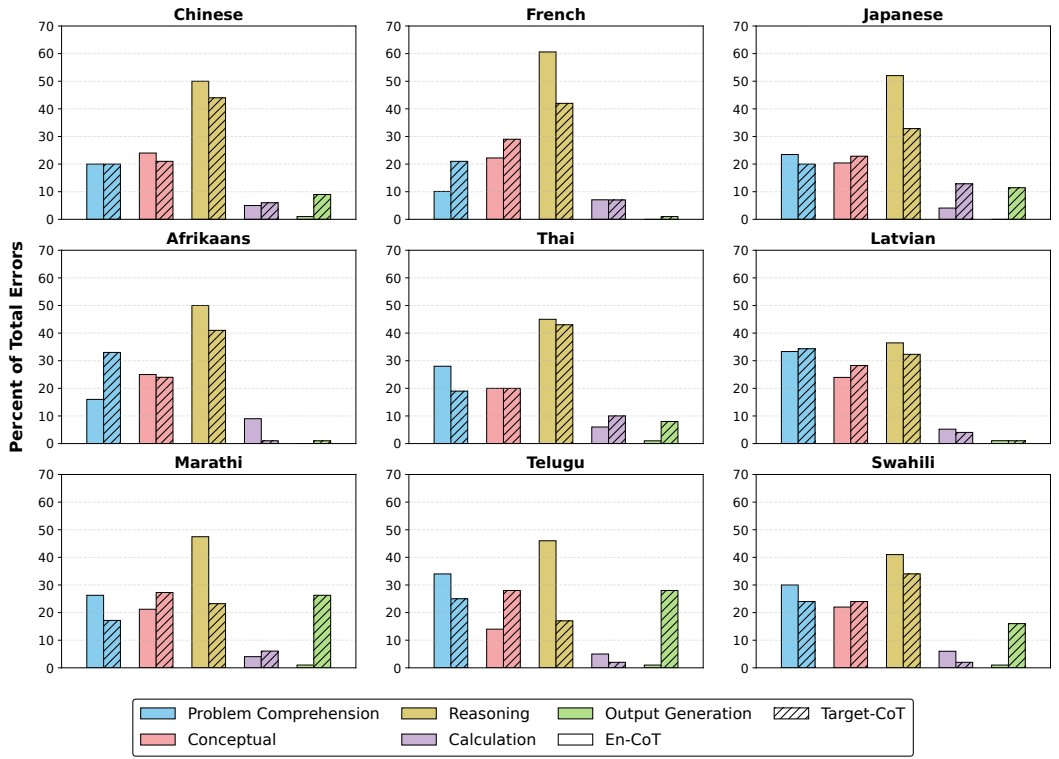

Figure 4: **Distribution of error types found in incorrect responses from DeepSeek-R1-Distill-Llama-70B on AIME-Combined.** The majority of errors in En-CoT stem from reasoning mistakes, while Target-CoT exhibits a higher proportion of output generation errors and conceptual errors compared to En-CoT.

from DeepSeek-R1-Distill-Llama-70B[3] on AIME-Combined. We leverage Gemini-2.5-Pro for error classification, with native speaker validation confirming accuracy (see Appendix A.9 for details). Figure 4 reveals a fundamental asymmetry in how errors manifest across reasoning settings. When models reason in English (En-CoT), nearly half of all errors (47.6%) stem from faulty reasoning steps, while conceptual misunderstandings account for 21.4% and output generation errors are negligible at 0.7%. In contrast, when reasoning occurs in the target language (Target-CoT), the error profile shifts markedly: reasoning errors decrease to 34.4%, while language-specific challenges become more prominent—output generation errors increase significantly to 11.3%, and conceptual errors rise slightly to 24.9%. Interestingly, problem comprehension and calculation errors remain relatively stable across both settings, indicating these error types are largely independent of the reasoning language choice.

Language-specific failure modes also emerge. Telugu's conceptual error rate doubles when reasoning in-language (28% of errors in Target-CoT vs 14% of errors in En-CoT), while Latvian shows persistently high comprehension errors regardless of reasoning language (about 34% of errors). These patterns suggest En-CoT and Target-CoT fail through fundamentally different mechanisms. En-CoT errors primarily reflect the inherent difficulty of reasoning, manifesting as logical failures once models engage with problems. Target-CoT, however, introduces additional barriers—unstable generation (e.g., endless repetition) in low-resource languages and difficulties applying concepts—that often prevent models from reaching the reasoning stage entirely.

## 8  CONCLUSION

This paper systematically investigates long CoT reasoning across nine non-English languages. We show that larger models do not necessarily lead to stronger multilingual reasoning, a 7B model reasoning in

---

[3]We chose this model as it had the most well-rounded performance across all tested languages.

English outperforming a 32B model reasoning in another language. We find that broad multilingual pretraining substantially improves both comprehension and reasoning; then, post-training on just 1,000 non-English language traces matches the performance of 20× more English traces for mid- and low-resource languages. Error analysis reveals fundamentally different failure modes: En-CoT primarily exhibits reasoning failures while Target-CoT suffers from language-specific generation errors and conceptual misunderstandings, particularly in low-resource languages. These findings show that developing models capable of seamless multilingual reasoning requires targeted interventions at each stage—cross-lingual transfer from English alone is insufficient.

ACKNOWLEDGMENTS

We would like to thank Syrielle Montariol and the NeuLab at CMU for valuable discussions.

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

# A EXPERIMENTAL DETAILS

## A.1 LANGUAGES

Table 3 lists the script, family, region, and approximate resource level of all the languages studied.

Table 3: Tested languages with their script, family, region, and approximate resource level.

| Language | Script | Language Family | Region | Resource Level |
|----------|--------|-----------------|--------|----------------|
| English | Latin | Indo-European (Germanic) | Global / Europe | High |
| Chinese | Han | Sino-Tibetan (Sinitic) | East Asia | High |
| French | Latin | Indo-European (Romance) | Europe / Africa | High |
| Japanese | Kanji + Kana | Japonic | East Asia | High |
| Afrikaans | Latin | Indo-European (Germanic) | Southern Africa | Mid |
| Thai | Thai | Kra–Dai | Southeast Asia | Mid |
| Latvian | Latin | Indo-European (Baltic) | Northern Europe | Mid |
| Marathi | Devanagari | Indo-European (Indo-Aryan) | South Asia | Low |
| Telugu | Telugu | Dravidian | South Asia | Low |
| Swahili | Latin | Niger–Congo (Bantu) | East Africa | Low |

## A.2 MODELS

We evaluate models across multiple scales and architectures to study cross-lingual long CoT reasoning. For scaling experiments (Section 4), we use the DeepSeek-R1-Distill family (1.5B, 7B, 14B, and 32B parameters), which share identical post-training on 800k English reasoning traces and derive from the Qwen 2.5 base family, allowing us to isolate parameter scaling effects. For pretraining experiments (Section 5), we compare four models: Qwen2.5-7B (18T tokens, 29 languages), Qwen2.5-Math-7B (Qwen2.5-7B with additional 1T+ math-focused tokens), Qwen3-8B-Base (36T tokens, 119 languages), and Gemma3-12B-PT (140 languages) enabling us to disentangle specialized reasoning pretraining from broad multilingual coverage. For post-training experiments (Section 6, Appendices B.4 and B.5), we fine-tune Qwen3-8B-Base and Olmo3-7B-Instruct. We choose the 7B/8B scale because it balances computational efficiency with reasoning capabilities for most experiments. To validate findings at larger scales (Appendix B.3), we evaluate DeepSeek-R1-Distill-Llama-70B and Llama-3.3-Nemotron-Super-49B, both based on Llama-3.3-70B-Instruct.

## A.3 TRAINING SETUP

Tables 4 and 5 provide the hyperparameter configurations for supervised fine-tuning (SFT) on OpenThoughts3-20k and Translated/Distilled-s1k, respectively. All models undergo full fine-tuning except Gemma3-12B-PT. For this model, we apply LoRA fine-tuning (Hu et al., 2022) with parameters detailed in Table 6 based on guidelines from Schulman & Thinking Machines Lab (2025). To enhance training efficiency, we employ the ZeRO optimizer (Rajbhandari et al., 2020) as implemented in the DeepSpeed library (Rasley et al., 2020). For SFT, we use the LLaMA-Factory (Zheng et al., 2024) framework and four NVIDIA L40S GPUs.

## A.4 LANGUAGE FORCING.

Controlling $L_{\text{reason}}$ during inference is difficult for reasoning models that have been heavily post-trained on English traces, particularly for low-resource languages. To address this, we employ language forcing (Yong et al., 2025), a prompting technique that encourages LLMs to reason in a specific target language. Language forcing works by injecting translated phrases and system prompts at the beginning of the generation process to guide the model's reasoning language. For example, to elicit French reasoning, we would seed the model's generation with *D'accord, je vais essayer de trouver une solution* (English: *Okay, I will try to find a solution*).

Table 4: SFT hyperparameters for OpenThoughts3-20k.

| Global Batch Size | Context Length | Learning Rate | Epochs |
|---|---|---|---|
| 64 | 32,768 | 1e-5 | 3 |

Table 5: SFT hyperparameters for Translated-s1k and Distilled-s1k.

| Global Batch Size | Context Length | Learning Rate | Epochs |
|---|---|---|---|
| 32 | 32,768 | 1e-5 | 5 |

Table 6: LoRA hyperparameters for Gemma3-12B-PT.

| Target Modules | Rank | Alpha | Dropout | Learning Rate |
|---|---|---|---|---|
| All | 64 | 128 | 0.05 | 1e-4 |

## A.5 EVALUATION SETUP

We evaluate on MATH-500, AIME-Combined, and MMLU-ProX using identical decoding settings (temperature 0.6, top-p 0.95, max output 32,768 tokens) and language forcing (defined in Appendix A.4) to control the reasoning language. We count a generation as correct only if the extracted final answer (e.g., the contents of `\boxed{·}`) matches the gold label and the response is language-compliant per Appendix A.7 (at least 90% target-language chunks). We mark non-compliant outputs as incorrect. For MATH-500 and MMLU-ProX, we decode one sample per item and report pass@1. For AIME-Combined, we decode five independent samples per problem (60×5 = 300) and report the mean correctness over all 300 generations (not best-of-k) to get more robust performance estimates.

## A.6 PROMPTS

The prompts used for translation and error analysis are presented in Figures 7 and 8, respectively.

## A.7 LANGUAGE COMPLIANCE ALGORITHM

Our language compliance algorithm validates whether a text response is written in a specified target language through a multi-stage processing pipeline. First, we remove LaTeX mathematical expressions, code blocks, symbols, and formatting while preserving natural language content across multiple scripts (Latin, CJK, Thai, Devanagari, Telugu). The cleaned text is then divided into fixed-size character chunks (128 characters), with duplicates removed. Each chunk undergoes language classification using either a script-ratio heuristic for non-Latin languages (checking if at least 75% of letters belong to the target script) or the Lingua statistical language detector (Stahl, 2025). We discard chunks where the language classification confidence is below 90%. The algorithm computes a compliance score as the ratio of target-language chunks to total valid chunks, requiring at least 90% of chunks to be in the target language for the response to be considered compliant.

## A.8 ERROR TAXONOMY

We develop a taxonomy comprising five distinct error categories that capture the primary failure modes observed in reasoning tasks. Table 7 presents these categories along with their definitions and representative examples.

## A.9 ERROR LABELING

To identify error patterns across languages, we randomly sample 100 incorrect, language-compliant responses for each language and setting (En-CoT, Target-CoT) from DeepSeek-R1-Distill-Llama-70B's outputs on AIME-Combined. We focus specifically on language-compliant responses to isolate reasoning

Table 7: **Taxonomy of error categories in reasoning.** Each category represents a distinct failure mode with specific characteristics and manifestations.

| Category | Definition | Example |
|---|---|---|
| Problem Comprehension | Misinterprets the objective, constraints, variables, or key terms. | Asked for *perimeter*, computes *area*. |
| Conceptual | Lacks/misuses required fact, formula, theorem, or principle. | Uses circle *area* formula to find *circumference*. |
| Reasoning | Flawed logical step or invalid inference; incorrect plan built from correct concepts. | From "If $A$ then $B$," concludes "If $B$ then $A$." |
| Calculation | Arithmetic/symbolic mistake despite a correct setup and plan. | Sets $x = 3 \times 7$ but computes 22. |
| Output Generation | Output violates constraints (format) or degenerates (stability). | Missing boxed output; repetitively generates "The next step is." |

errors from language compliance issues (see Appendix B.1 for compliance rates). For error classification, we prompt Gemini-2.5-Pro to categorize each response according to our five-category taxonomy, assigning a single error label based on the most critical mistake that leads to the incorrect answer (prompt shown in Figure 8). To validate this approach, native speakers manually review 20 samples each for Chinese, Telugu, French, and Japanese, confirming high agreement with the automated classifications.

# B ADDITIONAL RESULTS

## B.1 LANGUAGE COMPLIANCE RATES ACROSS MODELS

Because our evaluation criteria require models to generate responses that are both correct and language-compliant (Appendix A.7), results can conflate genuine target-language reasoning capabilities with a model's ability to follow instructions to reason in the target language. Table 8 presents language compliance rates for all models studied. We observe perfect language compliance when models reason in English (En-Only and En-CoT settings), so we omit those results. Overall, language compliance rates are consistently high across models and languages, with a few exceptions. Qwen2.5-Math-7B, which has undergone specialized continual pretraining on mathematical reasoning data, shows poor compliance in French, Afrikaans, and Swahili. Manual review reveals that model outputs typically begin with a few coherent sentences in the target language before switching to English. Notably, even after switching to English reasoning, the model's accuracy remains extremely low—15.6% for French, 13.6% for Afrikaans, and 5% for Swahili—far below its En-CoT performance. We also observe low compliance rates for DeepSeek-R1-Distill-Qwen-1.5B and DeepSeek-R1-Distill-Llama-70B when reasoning in Japanese. In both cases, manual review reveals frequent code-switching to Chinese. As with Qwen2.5-Math-7B, these models predominantly produce incorrect answers after switching languages, suggesting that the language switch does not improve their reasoning performance.

## B.2 THE EFFECT OF REASONING CHAIN LENGTH ON PERFORMANCE GAPS

In Section 4, we observe that performance gaps in long CoT are influenced more by the language used for reasoning than by the language of the input. This result contrasts with prior findings on short CoT, which report input comprehension as the primary obstacle for models (Ko et al., 2025). To explore this difference further, we use MMLU-ProX as a proxy for short CoT, as its questions elicit 2–4× fewer reasoning tokens than AIME-Combined (Table 9). Figure 5 shows that target-language reasoning performs competitively with English reasoning for high- and mid-resource languages on MMLU-ProX. Notably, target-language reasoning even *outperforms* English reasoning in Chinese and French at 32B parameters. This finding contrasts with the clear performance gap we observe on AIME-Combined (Figure 2). The most natural interpretation is chain length: when the chain is short, scaling mainly needs to recover input comprehension and the gap narrows; when the chain is long, sustained reasoning in the target language remains brittle and the gap widens. This pattern supports the result of Ko et al. (2025), where performance in short-CoT settings is limited more by comprehension than by the language used for reasoning.

Table 8: **Language compliance rates (%) across all models.** Percentage of responses that comply with language forcing instructions in the Target-CoT setting on AIME-Combined. Most models achieve high compliance rates across languages, with a few exceptions in certain model-language combinations.

| Models | EN | ZH | FR | JA | AF | TH | LV | MR | TE | SW |
|---|---|---|---|---|---|---|---|---|---|---|
| *Off-the-shelf Models* | | | | | | | | | | |
| DeepSeek-R1-Distill-Qwen-1.5B | 100 | 93.3 | 83.7 | 36.7 | 91.3 | 66.0 | 92.7 | 98.3 | 94.3 | 82.0 |
| DeepSeek-R1-Distill-Qwen-7B | 100 | 100 | 93.0 | 63.3 | 89.7 | 53.7 | 90.7 | 96.0 | 89.0 | 87.7 |
| DeepSeek-R1-Distill-Qwen-14B | 100 | 100 | 94.3 | 98.3 | 65.7 | 89.7 | 94.3 | 96.3 | 92.0 | 99.3 |
| DeepSeek-R1-Distill-Qwen-32B | 100 | 100 | 95.0 | 97.3 | 98.7 | 97.7 | 98.0 | 99.3 | 99.7 | 98.3 |
| DeepSeek-R1-Distill-Llama-70B | 100 | 94.0 | 87.7 | 25.3 | 84.0 | 81.3 | 94.0 | 98.0 | 100 | 99.3 |
| Llama-3.3-Nemotron-Super-49B | 100 | 92.0 | 86.0 | 97.3 | 73.7 | 98.7 | 95.7 | 94.0 | 98.3 | 94.7 |
| Qwen3-8B (Thinking) | 100 | 85.3 | 78.7 | 97.0 | 86.3 | 100 | 98.7 | 99.7 | 100 | 98.0 |
| *Fine-tuned on OpenThoughts3-20k* | | | | | | | | | | |
| Qwen2.5-7B | 100 | 100 | 44.7 | 93.7 | 58.0 | 91.3 | 99.0 | 90.3 | 95.0 | 98.3 |
| Qwen2.5-Math-7B | 100 | 71.7 | 34.0 | 67.7 | 29.3 | 94.0 | 90.7 | 75.3 | 94.3 | 34.0 |
| Qwen3-8B-Base | 100 | 91.0 | 97.7 | 100 | 93.3 | 100 | 94.3 | 83.7 | 69.0 | 88.7 |
| Gemma3-12B-PT | 100 | 84.7 | 97.7 | 95.0 | 95.3 | 90.7 | 97.0 | 85.7 | 88.0 | 97.7 |
| *Fine-tuned on s1k Variants* | | | | | | | | | | |
| English-s1k | 100 | 70.0 | 70.7 | 91.3 | 72.0 | 78.7 | 72.0 | 78.3 | 90.3 | 76.3 |
| Translated-s1k | 100 | 92.7 | 100 | 100 | 99.7 | 100 | 100 | 100 | 100 | 100 |
| Distilled-s1k | 100 | 94.7 | 100 | 100 | 100 | 99.7 | 100 | 99.7 | 100 | 100 |

Table 9: **Average response length (in thousands of tokens) by language and benchmark for DeepSeek-R1-Distill-Qwen-32B.** Responses on AIME-Combined are consistently 2-4× longer than on MMLU-ProX.

| Benchmark | EN | ZH | FR | JA | AF | TH | LV | MR | TE | SW | AVG |
|---|---|---|---|---|---|---|---|---|---|---|---|
| *En-CoT* | | | | | | | | | | | |
| • AIME-Combined | 12.1 | 11.0 | 11.0 | 11.2 | 11.6 | 11.1 | 11.1 | 11.1 | 10.7 | 11.7 | 11.3 |
| • MMLU-ProX | 3.1 | 2.8 | 3.1 | 2.8 | 2.9 | 2.4 | – | 2.9 | 3.0 | 3.4 | 2.9 |
| *Target-CoT* | | | | | | | | | | | |
| • AIME-Combined | 12.1 | 15.7 | 7.7 | 19.0 | 16.0 | 18.9 | 20.7 | 28.3 | 31.1 | 23.0 | 19.3 |
| • MMLU-ProX | 3.1 | 4.1 | 1.7 | 5.2 | 3.6 | 5.6 | – | 17.3 | 26.7 | 8.5 | 8.4 |

## B.3 VALIDATION OF SCALING TRENDS ON LARGER MODELS AND DIFFERENT ARCHITECTURES

In Section 4, we examine the DeepSeek-R1-Distill family of models ranging from 1.5B to 32B parameters. This family provides an ideal testbed for isolating scaling trends, as all models share the same post-training procedure and a common pretrained backbone. To validate these findings at larger scales and across different architectures, we extend our evaluation to two additional models: DeepSeek-R1-Distill-Llama-70B and Llama-3.3-Nemotron-Super-49B (Bercovich et al., 2025). Both models use Llama-3.3-70B-Instruct—a model "optimized for multilingual dialogue use cases" (Meta LLaMA Team, 2025)—as their reference model. As shown in Table 10, we observe the same substantial performance gap between target-language reasoning and English reasoning on AIME-Combined that we identify in Section 4. On average, the gap reaches 28.4% for DeepSeek-R1-Distill-Llama-70B and 40.8% for Llama-3.3-Nemotron-Super-49B. These results confirm that the performance disparity persists across both larger model scales and different base architectures.

## B.4 COMPARING ENGLISH AND TARGET-LANGUAGE DATA UNDER FIXED COMPUTE

In Section 6, we compared abundant English reasoning traces (e.g., OpenThoughts3-20k) with much smaller target-language datasets to examine the tradeoff between cross-lingual transfer and targeted supervision. However, this comparison conflates two factors: differences in data distributions (OpenThoughts3 vs. s1k) and, more critically, differences in fine-tuning compute. To isolate the effect of target-language training, we conducted a controlled experiment where all models were fine-tuned on exactly 1,000 traces. We generated synthetic datasets from the same English reasoning traces to ensure consistent data distribution. Table 11 compares English-s1k (baseline) against Translated-s1k and Distilled-s1k variants. This comparison reveals

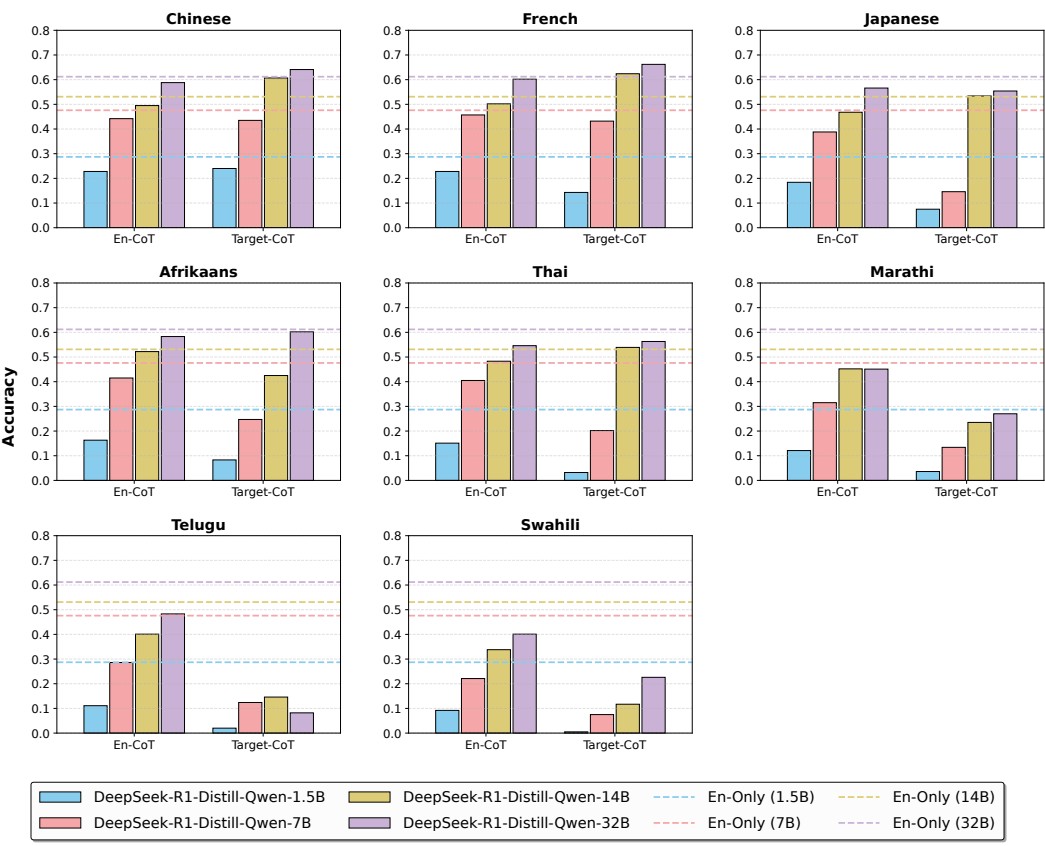

Figure 5: **Evaluation of scaling trends in DeepSeek-R1-Distill models on MMLU-ProX.** DeepSeek-R1-Distill models demonstrate narrower gaps between Target-CoT and En-CoT on short CoT tasks, with target-language reasoning in Chinese and French even outperforming English at multiple scales.

Table 10: **Large reasoning models exhibit persistent performance gaps between En-CoT and Target-CoT on AIME-Combined.** AVG excludes English, and EPR (English Performance Recovered) is computed as AVG / EN (%).

| Model | EN | ZH | FR | JA | AF | TH | LV | MR | TE | SW | AVG | EPR |
|---|---|---|---|---|---|---|---|---|---|---|---|---|
| Llama-3.3-Nemotron-Super-49B | | | | | | | | | | | | |
| • En-CoT | 69.7 | 70.7 | 69.7 | 68.7 | 67.7 | 66.0 | 61.0 | 63.3 | 56.7 | 65.7 | 65.5 | 94.0 |
| • Target-CoT | 69.7 | 41.0 | 47.3 | 28.0 | 36.7 | 20.3 | 12.3 | 15.3 | 9.3 | 12.0 | 24.7 | 35.4 |
| DeepSeek-R1-Distill-Llama-70B | | | | | | | | | | | | |
| • En-CoT | 56.3 | 58.0 | 58.7 | 55.0 | 52.3 | 51.0 | 47.7 | 50.0 | 45.3 | 55.3 | 52.6 | 93.4 |
| • Target-CoT | 56.3 | 39.7 | 41.7 | 6.0 | 30.3 | 21.3 | 21.3 | 14.7 | 17.3 | 25.3 | 24.2 | 43.0 |

that target-language training yields dramatic improvements across all benchmarks. Both Translated-s1k and Distilled-s1k substantially outperform English-s1k on non-English languages, with Translated-s1k achieving average gains of 44.9% on MATH-500, 12.8% on AIME-Combined, and 21.2% on MMLU-ProX. These results demonstrate that the apparent advantage of English-only training for high-resource languages in Section 6 was driven primarily by its 20× data advantage rather than effective cross-lingual transfer. When compute is held constant, target-language synthetic data is optimal across all resource levels.

## B.5 VALIDATION OF POST-TRAINING RESULTS ON OLMO MODELS

In Section 6 and Appendix B.4, our synthetic data experiments use Qwen3-8B-Base as the base model for post-training. To test whether these findings generalize beyond Qwen, we replicate the compute-equivalent setup with the open-source Olmo3-7B-Instruct Olmo Team et al. (2025) as the base model. Table 12 shows

Table 11: **Evaluation after fine-tuning Qwen3-8B-Base across compute-equivalent synthetic datasets (Target-CoT setting)**. Subscripts are computed relative to English-s1k (compute-equivalent baseline). The AVG excludes English and ignores missing entries (–). Both Translated-s1k and Distilled-s1k substantially outperform English-s1k across non-English languages, with Translated-s1k achieving the largest gains.

| SFT Dataset | EN | ZH | FR | JA | AF | TH | LV | MR | TE | SW | AVG |
|---|---|---|---|---|---|---|---|---|---|---|---|
| **MATH-500** | | | | | | | | | | | |
| English-s1k | 92.6 | 42.6 | 32.4 | 66.0 | 34.8 | 33.8 | 33.2 | 24.2 | 21.0 | 13.0 | 33.4 |
| Translated-s1k | $92.6_{+0.0}$ | $87.2_{+44.6}$ | $87.0_{+54.6}$ | $70.0_{+4.0}$ | $87.8_{+53.0}$ | $81.6_{+47.8}$ | $83.2_{+50.0}$ | $70.8_{+46.6}$ | $72.4_{+51.4}$ | $64.4_{+51.4}$ | 78.3 |
| Distilled-s1k | $92.6_{+0.0}$ | $60.0_{+17.4}$ | $78.8_{+46.4}$ | $64.4_{-1.6}$ | $85.4_{+50.6}$ | $83.8_{+50.0}$ | $82.0_{+48.8}$ | $76.2_{+52.0}$ | $67.2_{+46.2}$ | $57.8_{+44.8}$ | 72.8 |
| **AIME-Combined** | | | | | | | | | | | |
| English-s1k | 33.0 | 12.7 | 13.3 | 10.3 | 10.3 | 8.3 | 8.7 | 2.7 | 4.0 | 1.3 | 8.0 |
| Translated-s1k | $33.0_{+0.0}$ | $29.3_{+16.6}$ | $29.0_{+15.7}$ | $16.0_{+5.7}$ | $27.7_{+17.4}$ | $19.3_{+11.0}$ | $24.7_{+16.0}$ | $14.7_{+12.0}$ | $15.3_{+11.3}$ | $11.3_{+10.0}$ | 20.8 |
| Distilled-s1k | $33.0_{+0.0}$ | $10.7_{-2.0}$ | $21.0_{+7.7}$ | $13.0_{+2.7}$ | $28.3_{+18.0}$ | $19.7_{+11.4}$ | $22.0_{+13.3}$ | $16.3_{+13.6}$ | $13.7_{+9.7}$ | $12.7_{+11.4}$ | 17.5 |
| **MMLU-ProX** | | | | | | | | | | | |
| English-s1k | 69.0 | 28.9 | 31.0 | 45.6 | 23.6 | 29.3 | – | 14.6 | 12.8 | 8.2 | 24.3 |
| Translated-s1k | $69.0_{+0.0}$ | $58.7_{+29.8}$ | $59.5_{+28.5}$ | $41.2_{-4.4}$ | $55.1_{+31.5}$ | $49.1_{+19.8}$ | – | $40.0_{+25.4}$ | $34.0_{+21.2}$ | $26.5_{+18.3}$ | 45.5 |
| Distilled-s1k | $69.0_{+0.0}$ | $32.0_{+3.1}$ | $48.0_{+17.0}$ | $38.8_{-6.8}$ | $53.2_{+29.6}$ | $48.5_{+19.2}$ | – | $41.7_{+27.1}$ | $35.9_{+23.1}$ | $20.4_{+12.2}$ | 39.8 |

Table 12: **Evaluation after fine-tuning Olmo3-7B-Instruct across compute-equivalent synthetic datasets (Target-CoT setting)**. Subscripts are computed relative to English-s1k (compute-equivalent baseline). The AVG excludes English and ignores missing entries (–). Results show trends consistent with those observed using Qwen3-8B-Base as the base model.

| SFT Dataset | EN | ZH | FR | JA | AF | TH | LV | MR | TE | SW | AVG |
|---|---|---|---|---|---|---|---|---|---|---|---|
| **MATH-500** | | | | | | | | | | | |
| English-s1k | 91.8 | 64.6 | 76.8 | 44.2 | 40.8 | 17.0 | 15.4 | 25.2 | 16.8 | 14.4 | 35.0 |
| Translated-s1k | $91.8_{+0.0}$ | $76.0_{+11.4}$ | $79.8_{+3.0}$ | $57.4_{+13.2}$ | $69.6_{+28.8}$ | $41.6_{+24.6}$ | $48.8_{+33.4}$ | $30.6_{+5.4}$ | $16.0_{-0.8}$ | $43.8_{+29.4}$ | 51.5 |
| Distilled-s1k | $91.8_{+0.0}$ | $50.4_{-14.2}$ | $72.8_{-4.0}$ | $58.0_{+13.8}$ | $69.4_{+28.6}$ | $46.2_{+29.2}$ | $45.8_{+30.4}$ | $37.4_{+12.2}$ | $21.4_{+4.6}$ | $39.2_{+24.8}$ | 49.0 |
| **AIME-Combined** | | | | | | | | | | | |
| English-s1k | 33.7 | 11.7 | 16.7 | 3.3 | 3.0 | 0.0 | 1.0 | 0.0 | 0.0 | 1.3 | 4.1 |
| Translated-s1k | $33.7_{+0.0}$ | $16.7_{+5.0}$ | $20.0_{+3.3}$ | $7.3_{+4.0}$ | $13.0_{+10.0}$ | $6.3_{+6.3}$ | $7.0_{+6.0}$ | $2.7_{+2.7}$ | $1.3_{+1.3}$ | $9.0_{+7.7}$ | 9.3 |
| Distilled-s1k | $33.7_{+0.0}$ | $5.0_{-6.7}$ | $18.0_{+1.3}$ | $8.7_{+5.4}$ | $16.7_{+13.7}$ | $8.3_{+8.3}$ | $9.0_{+8.0}$ | $2.7_{+2.7}$ | $4.0_{+4.0}$ | $7.7_{+6.4}$ | 8.9 |
| **MMLU-ProX** | | | | | | | | | | | |
| English-s1k | 53.6 | 22.8 | 34.4 | 9.7 | 17.5 | 3.1 | – | 11.1 | 5.6 | 7.8 | 14.0 |
| Translated-s1k | $53.6_{+0.0}$ | $27.0_{+4.2}$ | $33.0_{-1.4}$ | $11.6_{+1.9}$ | $21.4_{+3.9}$ | $6.1_{+3.0}$ | – | $6.8_{-4.3}$ | $2.7_{-2.9}$ | $12.8_{+5.0}$ | 15.2 |
| Distilled-s1k | $53.6_{+0.0}$ | $16.0_{-6.8}$ | $28.4_{-6.0}$ | $13.6_{+3.9}$ | $25.9_{+8.4}$ | $7.5_{+4.4}$ | – | $7.8_{-3.3}$ | $2.7_{-2.9}$ | $8.3_{+0.5}$ | 13.8 |

similar trends: target-language training substantially outperforms English-only training under equivalent compute budgets.

Translated-s1k consistently outperforms English-s1k across most benchmarks and languages, with average gains of 16.5% on MATH-500, 5.2% on AIME-Combined, and 1.2% on MMLU-ProX. Although these gains are smaller than those observed with Qwen, the effect is directionally consistent: target-language synthetic data improves multilingual performance more than cross-lingual transfer from English at fixed compute. Distilled-s1k exhibits more variable behavior, particularly underperforming on high-resource languages such as Chinese and French on MATH-500 and MMLU-ProX, suggesting that the benefits of distillation depend on the base model's multilingual strengths.

## B.6 EVALUATION OF TRANSLATION CAPABILITIES

We first evaluate En→X translation quality on FLORES-200 devtest using spBLEU and chrF++ (Table 13). We compare Gemini-2.0-Flash (Google DeepMind, 2025), a general-purpose LLM, with NLLB-200-3.3B (Team et al., 2022) and MADLAD-400-10B (Kudugunta et al., 2023), which are dedicated MT systems

Table 13: **English-to-target translation performance on FLORES-200 devtest.** Comparison of Gemini-2.0-Flash (general-purpose LLM) against NLLB-200-3.3B and MADLAD-400-10B (dedicated MT systems). Scores reported as spBLEU / chrF++ with best scores in bold. Gemini-2.0-Flash achieves the highest performance across all languages, supporting its selection for generating Translated-s1k.

| En→X | Gemini-2.0-Flash | NLLB-200-3.3B | MADLAD-400-10B |
|---|---|---|---|
| ZH | **39.5 / 31.7** | 22.3 / 19.6 | 34.8 / 27.2 |
| FR | **59.2 / 72.1** | 53.5 / 68.0 | 55.7 / 69.9 |
| JA | **35.8 / 37.2** | 16.0 / 23.6 | 24.9 / 26.6 |
| AF | **45.9 / 64.9** | 43.0 / 63.6 | 44.0 / 64.5 |
| TH | **45.4 / 49.7** | 28.8 / 37.9 | 30.7 / 39.8 |
| LV | **43.5 / 61.1** | 27.1 / 48.9 | 38.3 / 57.2 |
| MR | **28.1 / 45.1** | 26.8 / 45.0 | 9.5 / 26.2 |
| TE | **40.4 / 55.2** | 36.5 / 51.9 | 28.3 / 46.4 |
| SW | **43.3 / 64.0** | 36.2 / 58.4 | 30.1 / 52.2 |

Table 14: **Reference-free quality estimation for paragraph-level translation of English reasoning traces.** We compare Gemini-2.0-Flash and NLLB-200-3.3B by translating English reasoning traces paragraph by paragraph and scoring each English–target pair with WMT23-COMETKiwi-DA-XL. Scores range from 0 (random translation) to 1 (perfect translation). We report the mean score over 100 randomly sampled traces for each language, with the best score in bold. MADLAD-400-10B is omitted because it could not reliably translate long paragraphs.

| En→X | Gemini-2.0-Flash | NLLB-200-3.3B |
|---|---|---|
| ZH | **0.619** | 0.473 |
| FR | **0.602** | 0.442 |
| JA | **0.632** | 0.404 |
| AF | **0.617** | 0.459 |
| TH | **0.617** | 0.490 |
| LV | **0.609** | 0.418 |
| MR | **0.526** | 0.384 |
| TE | **0.532** | 0.434 |
| SW | **0.576** | 0.390 |

covering roughly 200 and 400 languages, respectively. Gemini-2.0-Flash achieves the best scores on all evaluated languages, motivating its use to generate Translated-s1k.

However, FLORES-200 primarily contains general-domain prose and may not fully reflect the kinds of text we translate in Translated-s1k, such as mathematical notation and symbols. We therefore directly assess the quality of translations in Translated-s1k. Because we do not have human reference translations of the target-language reasoning traces, we use a reference-free quality estimation (QE) model, WMT23-COMETKiwi-DA-XL (Rei et al., 2023). We translate English reasoning traces paragraph by paragraph and score each English–target pair, reporting mean scores over 100 randomly sampled traces per language in Table 14. Across all languages, Gemini-2.0-Flash again outperforms the dedicated MT models, providing further support for our choice of translator.

## B.7 INFERENCE EFFICIENCY IN TERMS OF BYTES

We re-evaluate efficiency using UTF-8 bytes emitted rather than tokens (Figure 6). Across languages, the correlation between cost and accuracy becomes weakly negative indicating that cost is only loosely tied to performance when using a less biased measure of sequence length. These results suggest that script/tokenizer effects are primarily responsible for disparities in inference efficiency as opposed to overthinking.

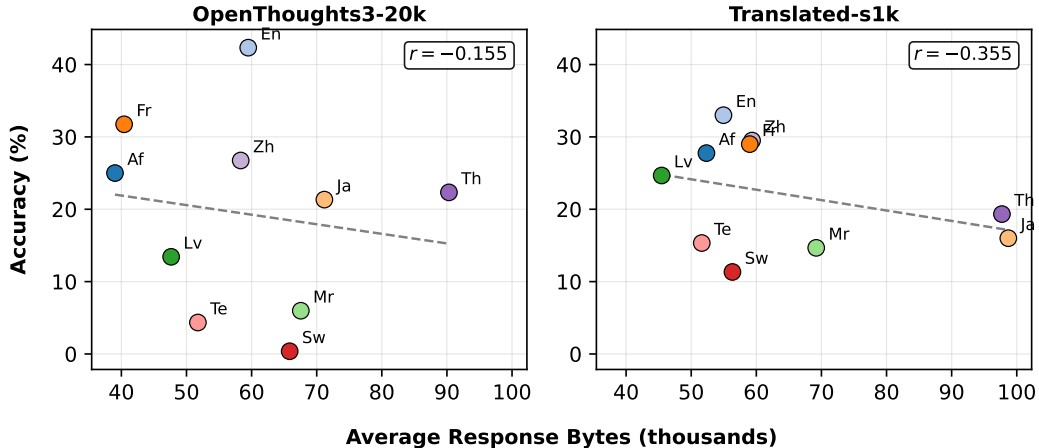

Figure 6: **Byte-based inference efficiency on AIME-Combined for fine-tuned Qwen3-8B-Base models.** Cost-accuracy correlations are weakly negative when measured in UTF-8 bytes rather than tokens.

```
Your job is to return a translated version of the English text.
* Translate to {target_lang}.
* The translation must be fluent, easy to read by native speakers.
* Do not solve the prompt translate it.
* You must preserve all details including math notations (latex) and
code.
* The math notations and code must not be translated, keep it as is.
* Return your translation in the following format.
<translation>
...translated text...
</translation>
The following is the source text for you task:
<English>
{source_text}
</English>
```

Figure 7: Prompt for translation.

```
You are an expert at identifying critical errors in model-generated responses. For each
case, you will receive:
1. A set of rubrics defining distinct errors.
2. A problem statement and a model-generated response that is incorrect.

**Your task:**
Identify which error, as defined by the rubrics, is the most critical in the model
response. You should only select the single error that was most influential in leading to
the model answer being incorrect. Your final answer must be a valid JSON object in the
format below, only one category should be present:
```json
{
"Problem Comprehension": {"present": 1|0, "explanation": "..."},
"Conceptual": ... ,
"Reasoning": ... ,
"Calculation": ... ,
"Output Generation": ...
}
```

---

Rubrics:
### 1. **Problem Comprehension Error**
This error occurs when the model **misunderstands what it's being asked to do**. The
reasoning can't be correct if it's aimed at the wrong goal.

* **Definition**: The model incorrectly interprets the problem's objective, constraints,
variables, or key terms.
* **Example**: Asked for the **perimeter** of a shape, the model calculates its **area**.

---

### 2. **Conceptual Error**
This error occurs when the model understands the goal but **lacks or misuses the
necessary knowledge** to devise a correct plan.

* **Definition**: The model fails to apply or incorrectly applies a required fact,
formula, theorem, or scientific principle.
* **Example**: Using the formula for a circle's area to find its circumference.

---

### 3. **Reasoning Error**
This is an error in the **logical flow** of the solution. The model understands the
problem and may know the right concepts, but it connects them incorrectly.

* **Definition**: The model employs a flawed logical step, makes an invalid inference,
or constructs an incorrect solution plan from correct concepts.
* **Example**: From "If A then B," incorrectly concluding "If B then A."

---

### 4. **Calculation Error**
This is a mechanical error in execution. The plan and logic are sound, but the math
is wrong.

* **Definition**: All prior steps are correct, but the model makes an error in an
arithmetic or symbolic computation.
* **Example**: Correctly setting up $x = 3 \times 7$ but calculating the result as 22.

---

### 5. **Output Generation Error**
This error covers any case where the model fails to deliver the output correctly.

* **Definition**: The model fails to produce the output according to the task's explicit
or implicit constraints. This includes issues with formatting (missing `\boxed{}`),
stability (repetitive loops), or adherence to instructions (wrong language).
* **Example**: Endlessly repeating "The next step is...".

Response to analyze:
{response}
End of response. Output your JSON object:
```

Figure 8: Prompt for error analysis.

