# OpenReview forum: "Long Chain-of-Thought Reasoning Across Languages"
_ICLR.cc/2026/Conference — ICLR 2026 Poster_

### Official Review · Reviewer_yQ3S · 2025-10-28

**Soundness:** 3
**Presentation:** 3
**Contribution:** 3
**Rating:** 6
**Confidence:** 4

**Summary:**

This paper systematically investigates long CoT reasoning in large language models across nine non-English languages. It distinguishes three reasoning settings—En-Only, En-CoT, and Target-CoT—and examines how multilingual reasoning evolves through model scaling, pretraining, post-training, and inference. The study finds that scaling enhances cross-lingual comprehension but fails to improve target-language reasoning; specialized reasoning pretraining can even degrade non-English performance, whereas broad multilingual pretraining benefits both. Post-training on translated reasoning traces outperforms distillation in target languages, especially for mid- and low-resource cases. Error analysis reveals distinct failure modes between English and non-English reasoning, underscoring that effective multilingual reasoning requires targeted interventions beyond simple cross-lingual transfer.

**Strengths:**

1. The paper conducts an extensive assessment of long chain-of-thought reasoning across nine non-English languages spanning high-, mid-, and low-resource settings, providing a solid empirical foundation for future multilingual reasoning research.

2. The study disentangles the effects of scaling, pretraining, post-training, and inference, offering clear and interpretable insights into how multilingual reasoning capabilities emerge and evolve.

3. The results reveal several unexpected yet important trends—such as specialized reasoning pretraining degrading target-language reasoning, and translated reasoning traces outperforming distilled ones—offering actionable implications for improving multilingual LLMs.

**Weaknesses:**

1. While the empirical results are comprehensive, the paper provides limited theoretical insight into why Target-CoT performance remains low even as models scale. A deeper analysis—e.g., examining the role of tokenizer design, cross-lingual alignment, or representational interference—would strengthen the interpretation of the observed trends.

2. The experiments primarily focus on mathematical reasoning tasks, which may understate the linguistic challenges inherent to multilingual CoT. Including broader reasoning or commonsense understanding benchmarks would make the conclusions more general and impactful.

3. The amount of English reasoning data greatly exceeds that of the target languages, which could introduce residual bias despite the normalization analysis in Appendix B.4. More controlled data scaling or balanced sampling experiments would help validate the robustness of the conclusions.

**Questions:**

1. **Line 112–116:** Could the authors clarify whether the trends observed for long chain-of-thought (Long CoT) reasoning are consistent with those for shorter CoT reasoning across languages? In particular, do the same cross-lingual gaps persist for shorter reasoning chains?

2. **Line 155:** The paper mentions using *Gemini 2.0 Flash* for translation. How do the authors ensure translation quality and linguistic diversity, especially across mid- and low-resource languages? Are there quantitative or human evaluations to validate translation accuracy?

3. **Section 4:** The scaling experiments primarily rely on long reasoning models such as DeepSeek-R1. Have the authors observed similar multilingual patterns in other model families (e.g., open-weight LLaMA or Qwen models) that are not explicitly optimized for long CoT reasoning?

4. **Line 258:** For high-resource non-English languages such as Chinese, Target-CoT performance drops even after scaling. Could this be partly due to bias introduced by the 20k English SFT data used during fine-tuning?

5. **Suggestion:** Including results from LLaMA-based models in the main paper (rather than the appendix) would strengthen cross-model comparisons and make the findings more broadly convincing.

---

---

> ### Author Response · Authors · 2025-11-21
>
> We thank Reviewer yQ3S for the thorough review. We appreciate your recognition of our "extensive assessment" of long CoT transfer, the "interpretable insights" from systematically disentangling development stages, and the "actionable implications" of our findings on specialized pretraining and translation-based approaches. Below, we address your questions and discuss how we have strengthened the paper accordingly:
>
> > “While the empirical results are comprehensive, the paper provides limited theoretical insight into why Target-CoT performance remains low even as models scale. A deeper analysis—e.g., examining the role of tokenizer design, cross-lingual alignment, or representational interference—would strengthen the interpretation of the observed trends.”
>
> We agree that a deeper theoretical understanding of why Target-CoT performance remains low is an important direction for future work. A detailed mechanistic analysis is beyond the scope of our current study, in part because long CoT traces make standard interpretability tools difficult to apply. Prior work [1,2] has focused on much simpler settings such as single-word translation, which are better suited to methods like activation patching and logit lens.
>
> That said, we believe our results provide a useful starting point. In Appendix B.2, we find that short CoT generalizes cross-lingually much more effectively than long CoT, even on tasks of comparable difficulty. This length-dependent divergence offers a concrete testbed for future representational or mechanistic analysis.
>
> [1] Do Llamas work in English? On the latent language of multilingual transformers
>
> [2] Separating Tongue from Thought: Activation Patching Reveals Language-Agnostic Concept Representations in Transformers
>
> > “The experiments primarily focus on mathematical reasoning tasks, which may understate the linguistic challenges inherent to multilingual CoT. Including broader reasoning or commonsense understanding benchmarks would make the conclusions more general and impactful.”
>
> In addition to math tasks, **Sections 4 and 6 include results on MMLU-ProX**, a multilingual, multi-disciplinary benchmark that tests knowledge-grounded reasoning beyond mathematics. These results show that the main trends we report also appear outside pure math (though often to a lesser extent). We intentionally exclude MMLU-ProX from Section 5 because Qwen2.5-Math-7B is explicitly optimized for mathematical reasoning, making a direct comparison on broader knowledge tasks misleading. In the revision, we clarify this design choice and more prominently reference the MMLU-ProX results when discussing our findings.
>
> > “The amount of English reasoning data greatly exceeds that of the target languages, which could introduce residual bias despite the normalization analysis in Appendix B.4. More controlled data scaling or balanced sampling experiments would help validate the robustness of the conclusions.”
>
> The controlled experiment in Appendix B.4 already accounts for differences in fine-tuning compute and data distribution. To further strengthen the robustness of our findings, we are currently re-running the experiments in Section 6 using the fully open-source Olmo3-7B as the base model instead of Qwen3-8B-Base. We will report the results here and add them to the revised paper (in Appendix B.5) when finished.
>
> > “Line 112–116: Could the authors clarify whether the trends observed for long chain-of-thought (Long CoT) reasoning are consistent with those for shorter CoT reasoning across languages? In particular, do the same cross-lingual gaps persist for shorter reasoning chains?”
>
> In **Appendix B.2**, we examine the cross-lingual gaps when models are evaluated on a task requiring shorter reasoning chains (MMLU-ProX). Importantly, we find that the gap between En-CoT and Target-CoT narrows significantly, with Target-CoT even outperforming En-CoT for high-resource languages like French and Chinese. This empirical observation raises theoretical questions about why short CoT generalizes better than long CoT.

---

> > ### Author Response · Authors · 2025-11-21
> >
> > > “Line 155: The paper mentions using Gemini 2.0 Flash for translation. How do the authors ensure translation quality and linguistic diversity, especially across mid- and low-resource languages? Are there quantitative or human evaluations to validate translation accuracy?”
> >
> > To ensure high quality translations, we evaluate Gemini 2.0 Flash against two strong MT baselines (MADLAD-400-10B-MT and NLLB-200-3.3B) on the FLORES-200 benchmark **(Table 13, Appendix B.6)**. Gemini 2.0 Flash attains the strongest automatic translation scores for all nine target languages, including the mid- and low-resource ones we study, which indicates that it produces high-quality and linguistically diverse translations for our setting.
> >
> > To bolster this claim, we also directly evaluate the quality of traces in Translated-s1k using a quality estimation model, WMT23-COMETKiwi-DA-XL. Gemini 2.0 again outperforms the MT models across all languages **(Table 14, Appendix B.6)**, further verifying the quality of the translations.
> >
> > > “Section 4: The scaling experiments primarily rely on long reasoning models such as DeepSeek-R1. Have the authors observed similar multilingual patterns in other model families (e.g., open-weight LLaMA or Qwen models) that are not explicitly optimized for long CoT reasoning?”
> >
> > To validate our findings in Section 4 at larger scales and across different architectures, we provide results on DeepSeek-R1-Distill-Llama-70B and Llama-3.3-Nemotron-Super-49B in **Appendix B.3**. Both models use Llama-3.3-70B-Instruct as their pretrained backbone, a model not explicitly optimized for long CoT reasoning. DeepSeek-R1-Distill-Llama-70B was fine-tuned using distillation from DeepSeek-R1, while Llama-3.3-Nemotron-Super-49B underwent a more flexible post-training process to enhance both its reasoning and non-reasoning capabilities. We observe the same trend as seen in Section 4: En-CoT recovers 93-94% of En-Only performance while Target-CoT recovers only 35-43%.
> >
> > > “Line 258: For high-resource non-English languages such as Chinese, Target-CoT performance drops even after scaling. Could this be partly due to bias introduced by the 20k English SFT data used during fine-tuning?”
> >
> > This explanation is plausible. Recent work [1] has found that Chinese shows particularly poor cross-lingual transfer with most languages, including English. This suggests that fine-tuning on English reasoning traces could introduce negative interference with existing Chinese reasoning capabilities. Negative interference from English data further motivates the need for target-language reasoning data which we explore in Section 6.
> >
> > [1] ATLAS: Adaptive Transfer Scaling Laws for Multilingual Pretraining, Finetuning, and Decoding the Curse of Multilinguality
> >
> > > “Suggestion: Including results from LLaMA-based models in the main paper (rather than the appendix) would strengthen cross-model comparisons and make the findings more broadly convincing.”
> >
> > Given the extra page for new results post-rebuttal, we will move the experiments with Llama models to the main paper. We would also like to acknowledge that we have added **new results with Gemma3-12B-PT to the main paper** that replicate our claims for multilingual pretraining. As mentioned above, we are also re-running experiments in Section 6 using Olmo3-7B as the base model. We selected Olmo3-7B instead of Llama3.1-8B because it is fully open-source rather than open-weights only, providing maximum transparency. These additions reduce the paper's focus on the Qwen family and provide stronger cross-model evidence for our findings.

---

> > > ### Comment · Reviewer_yQ3S · 2025-11-25
> > >
> > > Thank you for the thorough and detailed rebuttal. I am currently reviewing the additional analyses and clarifications provided, and I will finalize my assessment after completing this review.

---

### Official Review · Reviewer_SXHs · 2025-10-29

**Soundness:** 3
**Presentation:** 3
**Contribution:** 3
**Rating:** 6
**Confidence:** 4

**Summary:**

This paper presents a systematic study of long chain-of-thought (CoT) reasoning across nine non-English languages, examining how reasoning ability in large language models generalizes beyond English. The authors analyze four development stages—scaling, pretraining, post-training, and inference—and show that while model scaling and multilingual pretraining enhance comprehension of non-English inputs, reasoning within target languages lags far behind. Specialized reasoning pretraining (e.g., math-focused) improves English reasoning but often harms target-language reasoning. Fine-tuning with translated reasoning traces proves more effective than direct distillation, particularly for mid- and low-resource languages, and even small amounts of target-language data significantly improve performance. Error analysis reveals that English reasoning failures are mostly logical, whereas non-English reasoning is hindered by language-specific generation and conceptual errors.

**Strengths:**

- The paper tackles a novel and meaningful question—how long CoT reasoning transfers across languages—addressing a major gap in multilingual LLM research.

- Comprehensive evaluation across nine languages and multiple resource levels offers strong empirical grounding.

- The finding that translated synthetic data can substitute for large English datasets is practical and impactful for multilingual model training.

**Weaknesses:**

- The study relies solely on Qwen-family models (Qwen2.5, Qwen2.5-Math, Qwen3), limiting generalizability; results might be model-specific rather than universal.

- Although reports describe these Qwen models’ data composition, the exact pretraining and fine-tuning details remain opaque; using them as representative backbones may reduce methodological rigor.

**Questions:**

No.

---

> ### Author Response · Authors · 2025-11-21
>
> We thank Reviewer SXHs for their time and positive assessment. We appreciate your recognition of the novel research question, the breadth of our empirical evaluation across languages and resource levels, and the practical value of our findings on translated synthetic data. We understand your concerns about generalizability beyond the Qwen model family and the transparency of pretraining details. Below, we address these points:
>
> > “The study relies solely on Qwen-family models (Qwen2.5, Qwen2.5-Math, Qwen3), limiting generalizability; results might be model-specific rather than universal.”
>
> To address this, **we have extended each experimental result to include a different model family**, making our findings more generalizable. For Section 4, we already provide results on DeepSeek-R1-Distill-Llama-70B and Llama-3.3-Nemotron-Super-49B in **Appendix B.3**, demonstrating our findings across different scales, backbones, and post-training processes.
>
> For Section 5, we have added **new results with Gemma3-12B-PT** that replicate our claims for multilingual pretraining. Specifically, we claim that broad multilingual pretraining removes cross-lingual comprehension (En-CoT setting) as a bottleneck. Similarly to Qwen3-8B-Base, Gemma3-12B-PT recovers 85-97% of English performance in En-CoT, confirming this finding.
>
> For Section 6, we are currently re-running all experiments using the fully open-source Olmo3-7B as the base model instead of Qwen3-8B-Base. We will report the results here and add them to the revised paper **(in Appendix B.5)** when finished.
>
> > “Although reports describe these Qwen models’ data composition, the exact pretraining and fine-tuning details remain opaque; using them as representative backbones may reduce methodological rigor.”
>
> We acknowledge that the exact pretraining details of Qwen models are not fully transparent thus introducing potential confounds. As mentioned above, we extend our analysis in sections 4, 5, and 6 to other base models (Llama, Gemma, and Olmo) with different pretraining schemes/architectures, which demonstrates that the observed patterns are not tied to the peculiarities of Qwen models. For Sections 5 and 6, we perform the fine-tuning ourselves, so the post-training data and procedures are fully specified.

---

### Official Review · Reviewer_pui2 · 2025-10-29

**Soundness:** 2
**Presentation:** 2
**Contribution:** 3
**Rating:** 4
**Confidence:** 4

**Summary:**

This paper investigate the long-chain reasoning across languages, examining four stages of model development: scaling, pretraining, post-training, and inference. To achieve this, this paper compare three reasoning settings across nine non-English target languages: En-Only, En-Cot, and Target-Cot. The key findings are: (1) scaling reasoning model size improves multilingual task performance in En-CoT, but Target-CoT performance lags behind. (2) adding a specialized reasoning stage enhances En-CoT performance but degrades Target-CoT, whereas broad multilingual pretraining improves both modes simultaneously. (3) fine-tuning on reasoning traces automatically translated from gold English traces outperforms fine-tuning on target-language traces distilled from large reasoning models.

**Strengths:**

1. This paper first investigates the long cot in LLMs across languages.
2. The expriments offer  some insights for furture improvement.

**Weaknesses:**

1. The quality of the experiments remain to be improved.
2. Some experiment settings are strange.
3. The analyses could be more thorough.

All can refer to the questions below.

**Questions:**

1. In section 6, is it reasonable to translate English to low-resource language? How to ensure the translation quality?
2. Why conducting SFT in section 5? In section 5, the point is ``multi-lingual pre-training''.
3. In section 5, comparing Qwen-3 to Qwen-2.5 is not fair. Except the languages of pre-training data, the size, pre-training task, pre-training strategies are all different. Controlling variables for experiments is the optimal method.
4. In lines 181-182, I do not observe a ``consistently approach'' trand as model size scales. 7B LLMs with En-Cot are closest to corresponding En-Only baselines.
5. In section 6, comparing Translated-s1k and Distilled-s1k to OpenThoughts3-20k is unfair and unreasonable.
6. There are some issues of the format of Table 1.
7. This paper states many viewpoints in Introduction with only one citation. Although some of them are accepted by most researchers, citing their sources are very important.

---

> ### Author Response · Authors · 2025-11-21
>
> We thank Reviewer pui2 for their time and helpful feedback. We are pleased that you recognize the novelty of investigating long CoT reasoning across languages and the insights our experiments provide. We acknowledge your concerns about experimental design, fairness of comparisons, and the depth of our analyses. These are important points that we address comprehensively below:
>
> > “In section 6, is it reasonable to translate English to low-resource language? How to ensure the translation quality?”
>
> To ensure accurate translations, we evaluated Gemini 2.0 Flash against two strong machine translation baselines (MADLAD-400-10B-MT and NLLB-200-3.3B) using the FLORES-200 benchmark **(Table 13, Appendix B.6)**. Gemini 2.0 Flash achieved the highest spBLEU and chrF++ scores across all nine target languages, demonstrating its ability to produce high-quality translations even for lower-resource languages.
>
> We further validated translation quality using a second evaluation method. The revised paper includes new results **(Table 14, Appendix B.6)** that assess the Translated-s1K traces with WMT23-COMETKiwi-DA-XL, a quality estimation model. Gemini 2.0 again outperforms NLLB-200-3.3B across all languages, providing additional confirmation of the translation quality.
>
> > “Why conducting SFT in section 5? In section 5, the point is multi-lingual pre-training.”
>
> Section 5 is indeed about understanding the downstream impact of different pretraining strategies. However, base models cannot generate long CoT responses without post-training, which would prevent us from studying cross-lingual transfer of long CoT reasoning. Thus, we apply the same post-training procedure to all models in this section so that they are all able to generate outputs in the long CoT format. In this process, we SFT each model on the same 20k English traces from OpenThoughts3.  This results in post-trained versions of each model, where the models differ only by their pretraining data, but are also able to generate long CoT responses.
>
> > “In section 5, comparing Qwen-3 to Qwen-2.5 is not fair. Except the languages of pre-training data, the size, pre-training task, pre-training strategies are all different. Controlling variables for experiments is the optimal method.”
>
> Thank you for highlighting this limitation. We agree that the fairest comparison would control for model size and pretraining recipe while varying only multilingual coverage. Ideally, we would have performed continual pretraining on Qwen2.5-7B with expanded multilingual data, similar to the approach used for Qwen2.5-Math-7B. However, we lack the compute and data resources needed for such large-scale pretraining.
>
> Our main conclusions in Section 5 do not rely solely on Qwen3-8B-Base. The controlled comparison between Qwen2.5-7B and Qwen2.5-Math-7B already provides important evidence. In this comparison, model size and multilingual coverage are matched, with the primary change being additional English/Chinese reasoning pretraining. This comparison shows that strengthening English reasoning alone is not sufficient to improve Target-CoT and even degrades it.
>
> Building on this result, Qwen3-8B-Base provides complementary evidence. With broad multilingual coverage in pretraining, the gap between English and other languages is relatively small in the En-CoT setting but remains substantial for Target-CoT. Importantly, evaluating the gap between En-Only, En-CoT, and Target-CoT within the same model does not require cross-model comparison.
>
> Still, Qwen3-8B-Base is just one model, so we provide **new results with Gemma3-12B-PT**, another base model with broad multilingual coverage during pretraining (140 languages). We observe a consistent pattern: models with broad multilingual pretraining achieve higher EPR (English Performance Recovered), demonstrating more balanced reasoning capabilities. We present a summary of the results below, which can also be found in the revised paper:

---

> > ### Author Response · Authors · 2025-11-21
> >
> > **Table 2: Evaluation of each base model after identical English post-training. AVG is computed over all languages except English, and EPR (English Performance Recovered) is computed as AVG / EN (%), measuring cross-lingual transfer efficiency.**
> >
> > ### MATH-500
> >
> > **English CoT:**
> > | Base Model | EN | AVG | EPR |
> > |------------|-----|-----|-----|
> > | Qwen2.5-7B | 90.2 | 80.4 | 89.1 |
> > | Qwen2.5-Math-7B | 92.2 | 81.4 | 88.3 |
> > | Qwen3-8B-Base | 94.6 | 90.4 | 95.6 |
> > | Gemma3-12B-PT | 76.6 | 74.4 | **97.1** |
> >
> > **Target CoT:**
> > | Base Model | EN | AVG | EPR |
> > |------------|-----|-----|-----|
> > | Qwen2.5-7B | 90.2 | 32.3 | 35.8 |
> > | Qwen2.5-Math-7B | 92.2 | 12.6 | 13.7 |
> > | Qwen3-8B-Base | 94.6 | 62.0 | **65.6** |
> > | Gemma3-12B-PT | 76.6 | 39.7 | 51.8 |
> >
> > ### AIME-Combined
> >
> > **English CoT:**
> > | Base Model | EN | AVG | EPR |
> > |------------|-----|-----|-----|
> > | Qwen2.5-7B | 30.0 | 23.7 | 79.0 |
> > | Qwen2.5-Math-7B | 36.7 | 29.3 | 79.8 |
> > | Qwen3-8B-Base | 42.3 | 40.1 | **94.8** |
> > | Gemma3-12B-PT | 14.3 | 12.2 | 85.2 |
> >
> > **Target CoT:**
> > | Base Model | EN | AVG | EPR |
> > |------------|-----|-----|-----|
> > | Qwen2.5-7B | 30.0 | 6.0 | 20.0 |
> > | Qwen2.5-Math-7B | 36.7 | 1.7 | 4.6 |
> > | Qwen3-8B-Base | 42.3 | 15.9 | **37.6** |
> > | Gemma3-12B-PT | 14.3 | 5.1 | 35.5 |
> >
> > > “In lines 181-182, I do not observe a ``consistently approach'' trand as model size scales. 7B LLMs with En-Cot are closest to corresponding En-Only baselines.”
> >
> > Thank you for pointing this out. This statement was too vague. The patterns indeed vary by language in the En-CoT setting: For Chinese and French, all model sizes (1.5B, 7B, 14B, and 32B) perform on par with En-Only baselines. For Japanese, Afrikaans, Thai, and Latvian, only the 7B models match En-Only performance. For Marathi, Telugu, and Swahili, no model size approaches the En-Only baseline.
> >
> > We have revised lines 180-190 to discuss these results more precisely. However, we note that for high- and mid-resource languages, the 7B, 14B, and 32B models all remain roughly within 10% of the En-Only baseline, which represents a relatively small capability gap, especially compared to the substantial gap between Target-CoT and En-Only. This is why we remain optimistic about the effectiveness of scaling model capacity for improving En-CoT performance.
> >
> > > “In section 6, comparing Translated-s1k and Distilled-s1k to OpenThoughts3-20k is unfair and unreasonable.”
> >
> > In our original submission, we conducted a **controlled experiment in Appendix B.4** that is referenced in Section 6, where all models are fine-tuned on exactly 1k traces from s1k converted into different synthetic formats. The controlled results show that fine-tuning Qwen3-8B-Base on Translated-s1k yields substantial performance improvements over English-s1k : 44.9% on MATH-500, 12.8% on AIME-Combined, and 21.2% on MMLU-Pro.
> >
> > We maintain the Section 6 comparison because it reflects a practical trade-off that practitioners face: abundant English data versus smaller target-language datasets. We have clarified the writing to distinguish between the controlled study in Appendix B.4, which supports causal conclusions, and the Section 6 comparison, which provides practical guidelines.
> >
> > > “There are some issues of the format of Table 1.”
> >
> > We would appreciate it if you could describe the formatting issues further so that we can address them.
> >
> > > “This paper states many viewpoints in Introduction with only one citation. Although some of them are accepted by most researchers, citing their sources are very important.”
> >
> > Thanks for pointing this out, we have added citations in the introduction in the revised paper.

---

### Official Review · Reviewer_ZAqq · 2025-10-30

**Soundness:** 3
**Presentation:** 4
**Contribution:** 3
**Rating:** 6
**Confidence:** 4

**Summary:**

The paper investigates whether the "long chain-of-thought" abilities of large reasoning models actually transfer beyond English. It disentangles two evaluation modes across nine non-English languages. The first is En-CoT, where inputs are in the target language but the reasoning chain is in English, and the other is Target-CoT, where both input and reasoning are in the target language. Scaling improves multilingual performance primarily in En-CoT, while Target-CoT is left behind, especially on math tasks that require long multi-step chains. A controlled comparison reveals that math-specialized pretraining enhances En-CoT yet often harms Target-CoT, whereas broad multilingual pretraining improves both modes. For post-training, modest target-language supervision built by translating gold English traces generally outperforms distilling target-language traces from a strong teacher model. The paper also analyzes inference efficiency and error profiles, finding language-specific failure modes when reasoning in the target language.

**Strengths:**

The paper is well-written and framed clearly with three setups and a comprehensive evaluation of nine languages, covering high/middle/low resource languages.

Also, the scaling study is carefully controlled and highlights that Target-CoT never reaches English-reasoning levels, even at 32B; switching to target-language reasoning at 32B still performs lower than a 7B English baseline.

Besides, the post-training section is practical. It shows that with only ~1k target-language traces, translated from high-quality English rationales, models substantially outperform target-language distillation in aggregate and become comparable with much larger English-only SFT. It especially benefits mid/low-resource languages.

In addition, the analysis of inference efficiency shows that accuracy is negatively correlated with response length in tokens and that target-language SFT mitigates cross-lingual efficiency gaps. The byte-based view further investigates tokenizer effects.

Finally, the error analysis is quite insightful. Most of the errors in En-CoT are inference flaws, while the errors in Target-CoT are more reflected in output generation and conceptual levels. It also provides a qualitative example of a situation where Target-CoT fails while En-CoT succeeds.

**Weaknesses:**

Although the authors benchmark translators and justified the usage of  Gemini-2.0-Flash in Appendix B.5, it may still be promising to further measure the quality of the translated datasets with existing translation quality estimation metrics (e.g., xCOMET, MetricX). These scores will directly show that the translated datasets are reliable and trustworthy.

The evaluated model only covers one language family, that is, the Qwen series. Although Deepseek-Distilled-R1 is trained mainly on English and Chinese data, it still shows capability in multilingual reasoning. So I may suggest testing at least one Deepseek model and seeing if a similar phenomenon also happens there.

For the improvement part, the work only adopts supervised fine-tuning but does not try reinforcement learning, which has already become a popular strategy nowadays. Specifically, it may teach structured reasoning in non-English languages via verifiable rewards.

**Questions:**

Based on the weaknesses, I may post the following questions and suggestions:

(1) Adopt existing translation quality estimation metrics, like xCOMET, MetricX, to directly measure the translation quality.

(2) Evaluate with at least one DeepSeek series reasoning model, say, DeepSeek-R1-Distill-Qwen-7B or other preferences.

(3) Adopting RL post-training to see if it benefits Target-CoT, like using GRPO with a format reward of 0.2 if the language of the thinking traces matches the (low-resource) question language.

---

> ### Author Response · Authors · 2025-11-21
>
> We thank Reviewer Zaqq for the thoughtful review and constructive feedback. We appreciate your positive assessment of the paper's writing quality, the practical value of the post-training results, and the insights from the error analysis. Below, we address your comments:
>
> > “Adopt existing translation quality estimation metrics, like xCOMET, MetricX, to directly measure the translation quality.”
>
> We agree that translation quality estimation (QE) is important for establishing the reliability of our translated datasets. Since our English reasoning traces do not have human reference translations in the target languages, we use WMT23-COMETKiwi-DA-XL, a strong reference-free QE model, to score paragraph-level English–target pairs. For each language, we randomly sample 100 traces and report the mean score in **(Table 14, Appendix B.6)**. Scores range from 0 (random translation) to 1 (perfect translation). Gemini-2.0-Flash consistently achieves higher scores than NLLB-200-3.3B (a strong MT model) across all languages, supporting both our choice of translator and the reliability of the translated reasoning traces. MADLAD-400-10B is omitted because it could not reliably translate longer paragraphs. We have included the table below for reference:
>
> **Table 14. Reference-free quality estimation for paragraph-level translation of English reasoning traces.**
> | En→X | Gemini-2.0-Flash | NLLB-200-3.3B |
> |------|------------------|---------------|
> | ZH   | **0.619**        | 0.473         |
> | FR   | **0.602**        | 0.442         |
> | JA   | **0.632**        | 0.404         |
> | AF   | **0.617**        | 0.459         |
> | TH   | **0.617**        | 0.490         |
> | LV   | **0.609**        | 0.418         |
> | MR   | **0.526**        | 0.384         |
> | TE   | **0.532**        | 0.434         |
> | SW   | **0.576**        | 0.390         |
>
> > “Evaluate with at least one DeepSeek series reasoning model, say, DeepSeek-R1-Distill-Qwen-7B or other preferences.”
>
> We’d like to clarify that Section 4 already evaluates the DeepSeek-R1-Distill series from 1.5B to 32B. We additionally report results for DeepSeek-R1-Distill-Llama-70B and Llama-3.3-Nemotron-Super-49B in **Appendix B.3**, demonstrating that our findings generalize to different base architectures and scales. To further verify that the patterns we report are not specific to Qwen models, we have added new results with Gemma3-12B-PT in Section 5, and are currently re-running the experiments in Section 6 with Olmo3-7B instead of Qwen3-8B-Base as the base model (Appendix B.5).
>
> > “Adopting RL post-training to see if it benefits Target-CoT, like using GRPO with a format reward of 0.2 if the language of the thinking traces matches the (low-resource) question language.”
>
> Concurrent work [1,2] has proposed methods for multilingual RL while highlighting persistent challenges with cold-start SFT, reward shaping, and language consistency during RL. These studies focus on the RL component of post-training. In contrast, our work takes a broader perspective by studying long CoT transfer across all stages of model development, including but not limited to post-training.
>
> In Section 6, we examine post-training from the SFT perspective rather than the RL perspective. We prioritize SFT because it serves as an essential prerequisite to RL and there is a lack of existing research on synthetic approaches and resources for generating high-quality multilingual reasoning traces. Our finding that translated reasoning traces are highly sample efficient for Target-CoT, along with our released datasets, provide practical resources that can help researchers perform cold-start SFT before RL.
>
> [1] Aligning Multilingual Reasoning With Verifiable Semantics From A High-Resource Expert Model
>
> [2] When Verifiable Rewards Switch the Language: Cross-Lingual Collapse in Chain-of-Thought

---

### Author Response · Authors · 2025-12-02

We appreciate the thoughtful and positive reviews **(6, 4, 6, 6)**, with all four reviewers acknowledging strengths in our work. Reviewers recognized the **novelty of our contribution**, noting that our paper "*tackles a novel and meaningful question—how long CoT reasoning transfers across languages*" (SXHs) and is the "*first [to investigate] long cot in LLMs across languages*" (pui2). They also **praised our experimental setup**. The paper is "*framed clearly with three setups and a comprehensive evaluation of nine languages*" (ZAqq), providing an "*extensive assessment of long chain-of-thought reasoning across nine non-English languages*" (yQ3S) with "*comprehensive evaluation [that] offers strong empirical grounding*" (SXHs). Reviewers also found our **findings practically valuable**, highlighting that the "*post-training section is practical*" and will "*especially benefit mid/low-resource languages*" (ZAqq), while our synthetic data results are "*practical and impactful for multilingual model training*" (SXHs). Finally, they appreciated the **depth of our analysis**, calling our "*error analysis … quite insightful*" (ZAqq).

Two concerns that emerged across multiple reviewers were (1) validation of translation quality for the synthetic target-language traces, and (2) generalizability beyond the Qwen model family. We address both with three new results in the revision.

1. **Translation quality of Translated-s1k.**
   Several reviewers asked how we ensure that English→target-language translations are reliable, especially for low-resource languages (ZAqq, pui2, yQ3S). In response, we added two complementary evaluations:
   - **Standard MT benchmarks.** We already compare Gemini-2.0-Flash against MADLAD-400-10B-MT and NLLB-200-3.3B on FLORES-200 (Table 13, App. B.6) and show that Gemini-2.0-Flash consistently achieves the strongest spBLEU and chrF++ scores across all nine target languages.
   - **New QE on the actual reasoning traces.** To directly evaluate the Translated-s1k data, we now score paragraph-level English→X reasoning traces with WMT23-COMETKiwi-DA-XL, a strong reference-free quality estimation model (Table 14, App. B.6). Across all languages, Gemini-2.0-Flash achieves substantially higher COMETKiwi scores than NLLB-200-3.3B, indicating high-quality translations even for mid/low-resource languages. Both of these results directly support our claim that Translated-s1k is a reliable source of target-language reasoning supervision.

2. **Generalizability beyond Qwen models.**
   Multiple reviewers expressed concern that the main findings might be specific to Qwen models (ZAqq, SXHs, yQ3S). In the revised version, we systematically broaden our model coverage at all stages of analysis:
   - **Scaling (Section 4):** We clarify that we already report results for DeepSeek-R1-Distill-Llama-70B and Llama-3.3-Nemotron-Super-49B (App. B.3), both based on Llama-3.3-70B-Instruct. These models reproduce the same key pattern as the DeepSeek-R1-Distill series: scaling En-CoT recovers ~93–94% of En-Only performance, while Target-CoT recovers only ~35–43%. We have moved
   - **Multilingual pretraining (Section 5):** We add new experiments with Gemma3-12B-PT, a base model pretrained on 140 languages. After identical English long-CoT post-training, Gemma3-12B-PT shows high English Performance Recovered (EPR) in En-CoT, mirroring Qwen3-8B-Base and confirming that broad multilingual pretraining consistently removes cross-lingual *comprehension* as the main bottleneck.
   - **Post-training with synthetic data (Section 6):** We replicate our compute-equivalent post-training setup with the fully open-source Olmo3-7B-Instruct (App. B.5). The new Olmo results show the same trend: Translated-s1k consistently outperforms English-s1k under equivalent compute, with average gains of +16.5% on MATH-500, +5.2% on AIME-Combined, and +1.2% on MMLU-ProX, while Distilled-s1k has more variable behavior and underperforms on some high-resource languages.

In summary, these new translation-quality evaluations and cross-family experiments (Gemma3, Llama3, and Olmo3) directly address the main shared concerns and **improve the generality of our findings**. We respond to individual reviewer concerns in our comments below. Overall, we feel that the reviewers’ time and effort have improved the quality of our manuscript and we thank them for this.

---

### Meta-Review · Area_Chair_nAwP · 2025-12-24

**Summary:**

This paper presents an empirical study of CoT reasoning across nine non-English languages. The authors systematically evaluate the impact of scaling, pretraining, and post-training strategies, finding that translation-based synthetic data can be effective for multilingual alignment. Reviewers generally acknowledged the thoroughness of the experimental setup and the practical relevance of the findings. During the rebuttal, the authors addressed concerns regarding model dependence by extending their evaluation to include Llama, Gemma, and Olmo families. The extensive benchmarking and consistent empirical results support the paper's acceptance.

**Reviewer Concerns:**

Addressed:
1. Multiple reviewers questioned the quality of the translated reasoning traces (Translated-s1k), specifically for low-resource languages. The authors added a new Quality Estimation analysis using WMT23-COMETKiwi-DA-XL to score the actual reasoning traces. They demonstrated that their translation method consistently outperformed strong baselines across all target languages.
2. Multiple reviewers worried the findings were specific to the Qwen model family. The authors expanded their experiments to three additional model families: Llama-3, Gemma-3, Olmo-3.

Outstanding:
1. Reviewer yQ3S noted that while the empirical results are strong, the paper lacks theoretical explanation for the empirical observations. This is also my major concern about this paper.
2. Reviewer ZAqq suggested the current paper did not try RL, only tried SFT.
3. Reviewer yQ3S felt the focus on math might understate linguistic challenges and suggested broader reasoning benchmarks.

**Reviewer Scores:**

Reviewer ZAqq may not increase the score because he/she want the authors to try RL.

Reviewer pui2 may not increase the score because he/she has many concerns about the paper (weakness), but the questions did not fully cover all the weakness points. Therefore, it seems difficult to revert his opinion.

Reviewer SXHs may increase the score because his/her concerns are addressed during rebuttal.

Reviewer yQ3S may not increase the score because some of his/her concerns were not addressed during rebuttal, especially for the theoretical understanding one.

---

### Decision · Program_Chairs · 2026-01-26

Accept (Poster)